# Beyond Predefined Depots: A Dual-Mode Generative DRL Framework for Proactive Depot Generation in Location-Routing Problem

## Abstract

The Location-Routing Problem (LRP), which combines the challenges of facility (depot) locating and vehicle route planning, is critically constrained by the reliance on predefined depot candidates, limiting the solution space and potentially leading to suboptimal outcomes. Previous research on LRP without predefined depots is scant and predominantly relies on heuristic algorithms that iteratively attempt depot placements across a planar area. Such approaches lack the ability to proactively generate depot locations that meet specific geographic requirements, revealing a notable gap in current research landscape. To bridge this gap, we propose a data-driven generative DRL framework, designed to proactively generate depots for LRP without predefined depot candidates, solely based on customer requests data which include geographic and demand information. It can operate in two distinct modes: direct generation of exact depot locations, and the creation of a multivariate Gaussian distribution for flexible depots sampling. By extracting depots' geographic pattern from customer requests data, our approach can dynamically respond to logistical needs, identifying high-quality depot locations that further reduce total routing costs compared to traditional methods. Extensive experiments demonstrate that, for a same group of customer requests, compared with those depots identified through random attempts, our framework can proactively generate depots that lead to superior solution routes with lower routing cost. The implications of our framework potentially extend into real-world applications, particularly in emergency medical rescue and disaster relief logistics, where rapid establishment and adjustment of depot locations are paramount, showcasing its potential in addressing LRP for dynamic and unpredictable environments.

## 1 Introduction

The Location-Routing Problem (LRP) is a critical optimization challenge in the urban logistics industry, combining two interdependent decisions: selecting depot locations where vehicles commence and conclude their tasks, and planning vehicle routes for serving customers. This integration is crucial as the depot locations can directly affect the vehicle route planning, thereby impacting overall costs Salhi & Rand (1989). The LRP can be formally defined as Nagy & Salhi (2007): *Given a set of customers with specific location and quantity of demands, and a set of potential depot candidates each with a fleet of vehicles featuring fixed capacity, aiming to properly select a subset of depots and plan routes for vehicles departing from these chosen depots to meet customers' demands, while minimizing both depot-related and route-related costs, without violating specific constraints.*

In this traditional problem configuration, solving LRP have relied on a predefined set of depot candidates Contardo et al. (2014); Nguyen et al. (2012); Pourghader Chobar et al. (2022); Wang et al. (2023) instead of directly generating desired optimal depot locations, thereby limiting the solution space and potentially leading to suboptimal outcomes. This constraint is particularly pronounced in scenarios where the optimal depot locations are not included in the candidates set, or when the problem configuration demands a high degree of flexibility in depot placement, requiring quickly establish and adjust depot locations. The real-world application that underscores the necessity of generating depots without predefined candidates is medical rescue and disaster relief logistics: In the aftermath of a natural disaster, such as an earthquake or flood, the existing infrastructure may be

severely damaged, rendering previously established depots unusable. In such scenarios, the ability to dynamically generate new depot locations based on current needs and constraints is crucial for efficient and effective relief operations.

Regarding this extended LRP scenario without predefined depot candidates, only a limited number of studies do the exploration by considering the concept of an infinite candidates set, represented by a planar area, for depot selection. However, these works primarily focus on designing heuristic algorithms to iteratively initiate new locations as depots across the planar area, and only manage to consider up to 2 depots (single-depot work Schwardt & Dethloff (2005); Schwardt & Fischer (2009), double-depot work Salhi & Nagy (2009)). Furthermore, these attempts demonstrate low efficiency and adaptability in tackling specific location constraints for depots, such as the required specific distance range among depots. In this situation, if simply expanding the search of the depot candidates across the map and aimlessly attempting new points, then undesired increasing on problem scale will be incurred, thereby leading to excessive time consumption and expensive computation. Therefore, devising a method to proactively generate high-quality depots, satisfying the depot location constraints for LRP scenario without predefined depot candidates, is well-motivated.

Motivated by this necessity of proactively generating depots when no candidates are predefined, we propose a generative deep reinforcement learning (DRL) framework, uniquely crafted to address LRP in depot-generating fashion. By leveraging customers' logistical requests data, which encompass geographic locations and specific demands, our framework generates depot locations and plans efficient routes for vehicles dispatched from these generated depots for serving the customer requests. Specially, our framework encompasses two models: *(1) Depot Generative Model (DGM)*, a deep generative model capable of generating depots in two distinct modes: direct generation of exact depot locations *or* production of a multivariate Gaussian distribution for flexible depots sampling. The exact mode ensures precision when necessary, while the Gaussian mode introduces sampling variability, enhancing the model's generalization and robustness to diverse customer distributions. *(2) Multi-depot Location-Routing Attention Model (MDLRAM)*, an end-to-end DRL model focusing on providing an efficient LRP solution for serving customers based on the generated depots, with minimized objective including both route-related and depot-related cost.

In summary, the contributions of our work include: (1) A generative DRL framework for LRP that proactively generates depots based on customer requests data, eliminating the reliance on predefined depot candidates, with a particular emphasis on applications requiring rapid adaptability, such as disaster relief logistics; (2) The component model - DGM - provides two distinct operational modes for depot generation: direct generating exact depot locations and producing a multivariate Gaussian distribution for flexible depots sampling, catering to a diverse range of real-world scenarios; (3) The component model - MDLRAM - provides an integrated LRP solution, minimizing the objectives including both route-related and depot-related cost, while also offering flexibility to adjust inter-depot cost distribution for balanced cost management across multiple depots. (4) The detachability of our framework allows both independent or combined usage of its components. DGM's depot-generating ability can be fine-tuned to adapt to various LRP variants through integration with other models, while MDLRAM can be freely used in traditional LRP configuration with predefined depot candidates, and also can be fine-tuned to accommodate various real-life constraints.

## 1.1 RELATED WORK

**Methods for LRP with Predefined Depot Candidates:** In addressing the LRP with Predefined Depot Candidates, traditional methods have predominantly employed exact and heuristic approaches. Exact methods, such as Mixed Integer Programming (MIP) models enhanced by branch-and-cut Belenguer et al. (2011); Akca et al. (2009) or column generation techniques Contardo et al. (2014), offer precision but often struggle with scalability in larger and complex scenarios due to an exponential increase in binary variables. This limitation has pivoted attention towards heuristic methods, which are categorized into: matheuristic approaches Rath & Gutjahr (2014); Danach et al. (2019); Ghasemi et al. (2022) that blend heuristic rules with exact methods, learning-aided heuristics Prins et al. (2006); Nguyen et al. (2012) that leverage learning-based algorithms to refine heuristic operations, and pure meta-heuristic algorithms. Among pure meta-heuristics, cluster-based heuristics Billionnet et al. (2005); Barreto et al. (2007) and iterative methods Salhi & Nagy (2009); Pourghader Chobar et al. (2022); Albareda-Sambola et al. (2007) have been notable. However, the cluster-based heuristics, which focus on geographically clustering customers, exhibit limitations in handling additional

constraints like customer-specific time windows, while the iterative methods present insufficient conjugation between the two stages of depot-selecting and route-planning. Besides, these methods typically require initiating a new search process for each case, leading to inefficiencies when even minor alterations occur to current problem instance.

The advancements in DRL have shown promise in addressing routing problems, both in "learn-to-construct/generalize" Kool et al. (2019); Xin et al. (2021a); Lin et al. (2024); Zhou et al. (2024) and "learn-to-improve/decompose" Xin et al. (2021b); Ma et al. (2021); Ye et al. (2024). However, its application in LRP, which integrates the challenges of facility locating with routing problem, still remains notably underexplored due to their inherent limitations in problem formulation for scenario involving multiple depots and the inability to organically integrate depot-selecting with route-planning. The works Arishi & Krishnan (2023); Rabbanian et al. (2023); Anuar et al. (2021) focus on resolving routing problems involving multiple depots, but without considering the depot-relate cost, which technically confine them as multi-depot VRP, instead of LRP which considers both route-related cost and depot-related cost. The work Wang et al. (2023) considers depot-related cost but adopts a two-stage process, clustering customers with an assigned depot location first and planning routes second, thereby separating depot selection from route planning, which fails to capture the interdependencies between these two critical aspects, while also lacking verification on standard LRP setup align with real-world datasets. Most importantly, all these methods are constrained to the predefined depot candidates, falling short in dealing with LRP without predefined depot choices.

**Exploration of LRP without Predefined Depot Candidates:** Only a scant number of studies explore the LRP without predefined depot candidates, predominantly employing heuristic strategies for attempting new depots across a planar area devoid of predefined depot choices. The work Schwardt & Dethloff (2005); Schwardt & Fischer (2009) concentrate on single-depot scenario, where a single and uncapacitated depot is to be selected from a planar area. Specifically, Schwardt & Fischer (2009) extends the cluster-based method in Schwardt & Dethloff (2005), proposing a learning-aided heuristic method to recurrently initiate new depot. Based on the same single-depot scenario, the work Manzour-al Ajdad et al. (2012) proposes a hierarchical heuristic method to iteratively update candidate circle to select new depot and then plan routes based on this depot. Furthermore, Salhi & Nagy (2009) extends the iterative heuristic method to explore multi-depot scenario, but only manages to deal with cases with up to two depots.

It is notable that, compared with our method's endeavors on actively and directly generating the recommended depots, these works employ heuristic algorithms to iteratively attempt new depot and then decide if it is a better one by re-planning routes based on it, limited to single or double-depot scenarios. Moreover, all these works lack ability on considering specific location constraints for depots, highlighting the necessity for a more adaptable and flexible solution.

## 2 METHODOLOGY

**Overview: Chain-of-Thoughts**   Solely based on the customer requests data which include geographic and goods demand information within an area, in pursuit of a solution that proactively generates high-quality depots satisfying specific location requirements, and subsequently plans optimized vehicle routes from these generated depots to efficiently serve customers, we propose the generative DRL framework, which is overviewed in Fig. 1.

The Depot Generative Model (DGM) takes in the customer requests information including the positions and specific quantity of demands, generating the required amount of depots in two distinct modes: the exact depot locations or a multivariate Gaussian distribution for flexible depots sampling.

Training DGM for generating desired depots hinges on a robust evaluation mechanism to assess the quality of the generated depot set, i.e., based on the same group of customer requests and identical route-planning strategies, determining which set of depot can lead to solution routes with lower routing cost. This necessitates a *critic model* to score the generated depot locations or the distribution during training. Additionally, to facilitate an efficient training process, this critic model must be able to instantly provide scores for the generated depot set, and also has to operate batch-wisely.

In this pursuit, we modify the Attention Model Kool et al. (2019) to accommodate the multi-depot scenario, introducing the Multi-depot Location-Routing Attention Model (MDLRAM) as a critic model placed after the DGM, constituting the entire framework. By taking in the customer requests

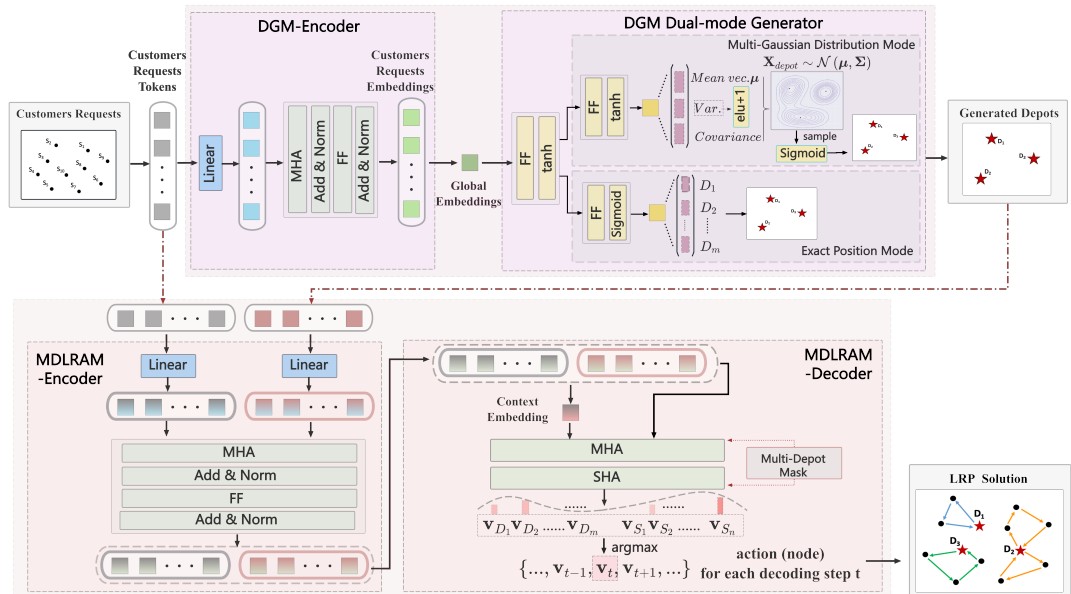

Figure 1: Overview of the Generative DRL framework for Depot Generation in LRP.

and the generated depots from DGM, MDLRAM outputs the LRP solution routes, associated with a minimized objective, ***providing score*** to rate the generated depot locations or the distribution.

Because MDLRAM serves as a critic model for DGM, it should be robust enough to provide a reliable score for assessing the generated depots from DGM. That means, the score is expected to solely reflect the quality of the generated depots, ruling out the influence of LRP routing solution's quality per se as much as possible. To achieve this capability, the MDLRAM should be ***pre-trained*** to be able to provide the LRP routing solution with minimized overall cost based on given requests and depots, and then set fixed to participate in the training of DGM. In this manner, during training the DGM, the different sets of depots generated by DGM for a same group of customer requests will get different scores from MDLRAM, solely reflecting the influence of depot locations, thereby facilitating a robust training process for DGM.

## 2.1 CRITIC MODEL: MDLRAM

As a critic model for DGM, the MDLRAM takes in the customer requests and the generated depots, aiming to output the integrated LRP routing solution with minimized objectives including both route-related and depot-related cost.

**MDLRAM-Configuration:** In alignment with the conventional setting Belenguer et al. (2011), the configuration is defined on an undirected graph $G = (V, E)$, where the $V = \{\mathbf{v}_{D_1}, \ldots, \mathbf{v}_{D_m}, \mathbf{v}_{S_1}, \ldots, \mathbf{v}_{S_n}\}$ denote the vertices set comprising $n$ customers and $m$ depots. Specifically, $\mathbf{v}_{D_k}$ signifies the coordinates $(x_{D_k}, y_{D_k})$ for depot $D_k$, where $k \in \{1, \ldots, m\}$, and $\mathbf{v}_{S_e}$ represents the coordinates $(x_{S_e}, y_{S_e})$ for customer $S_e$, where $e \in \{1, \ldots, n\}$. The Euclidean edge set is defined as $E \subseteq V \times V$, with $d_{ij}$ representing the Euclidean distance from $\mathbf{v}_i$ to $\mathbf{v}_j$.

Each customer $\mathbf{v}_{S_e}$ has a specific quantity of demands for goods denoted as $q_e$. Each depot $\mathbf{v}_{D_k}$ is characterized by two attributes: (i) the maximum supply $M_k$ (soft constraint), indicating the desired maximum total goods dispatched from this depot; (ii) the fixed opening cost $O_k$, indicating the expense for using this depot facility. Regarding the vehicles, we operate a homogeneous fleet, with each vehicle having the same maximum capacity $Q$ (hard constraint) indicating the maximum vehicle load during service, and a setup cost $U$ for using this vehicle in service. (More details for LRP configuration are available in Appendix A.1).

**MDLRAM-Objective Function:** In the LRP scenario, a feasible solution is essentially a set of routes, simultaneously executed by multiple vehicles starting and ending at their designated depots. To utilize DRL model to output solution routes, it's crucial to formulate the solution routes into a Markov Decision Process (MDP) as the output format, representing iterative decisions to construct

the solution. To mathematically formulate the LRP solution into MDP, a tuple $(\mathbf{S}, \mathbf{A}, \mathbf{P}, \mathbf{R}, \gamma)$ is defined, with each decision step $t$ associated with a tuple $(s_t, a_t, p_t, r_t, \gamma_t)$. The $s_t$ represent current state, encompassing information of: the current route's depot, the current serving customer and remaining capacity on current vehicle; The action $a_t$ denotes the next visit point, subject to the vehicle's remaining load; The $p_t$ and the $r_t$ correspond to the transition probability and cost associated with action $a_t$, respectively. Along with the generation of MDP, in each decision step $t$, the current state is dynamically updated upon serving a customer or returning to a depot. (See Appendix A.2 for detailed MDP formulation proposed for LRP scenario.)

Following this construction, a feasible LRP solution is formulated, associated with an objective (cost) function expressed as Eq. (1). Apart from the step-wisely accumulated transit distance length $\sum_t r_t$ along with the MDP generating process, other costs, which depict solution's overall performance are also integrated into the total cost with a respective discount, including: *(i) the opening cost for used depots; (ii) the setup cost for dispatched vehicles; (iii) penalty of exceeding depot desired maximum supply*. In Eq. (1), $\eta_k \in \{0, 1\}$ represents whether depot $D_k$ is opened, $\chi_k$ records the number of vehicles dispatched from depot $D_k$, and $\alpha, \beta, \delta$ are coefficients.

$$L_{\text{Sel}}(\mathbf{A}) = \sum_t r_t + \alpha \cdot \sum_{k=1}^m O_k \cdot \eta_k + \beta \cdot \sum_{k=1}^m U \cdot \chi_k + \delta \cdot \sum_{k=1}^m \max[(\sum_e q_e)_k - M_k, 0] \quad (1)$$

MDLRAM aims to minimize the expectation of this loss associated with the LRP solution, defined as $\mathbb{E}[L_{\text{Sel}}(\mathbf{A})]$, where $L_{\text{Sel}}(\mathbf{A})$ is expressed in Eq. (1).

**MDLRAM-encoder:** As is shown in the red block of Fig. 1, two streams of information are fed into MDLRAM as input: the depot candidates and the customer requests data. For each depot candidate $\mathbf{v}_{D_k}$ where $k \in \{1, \ldots, m\}$, it is represented by its coordinates $\mathbf{g}_{D_k} = [x_{D_k}, y_{D_k}]^T$. For each customer $\mathbf{v}_{S_e}$ where $e \in \{1, \ldots, n\}$, it is depicted by a vector concatenating its coordinates and specific demands, in form of $\mathbf{g}_{S_e} = [x_{S_e}, y_{S_e}, q_e]^T$. By respectively implementing different learnable linear projections, these depot candidates information vectors and customers information vectors are embedded into a high-dimensional space with same dimension, deriving the node features $\{\mathbf{h}_{D_1}, \ldots, \mathbf{h}_{D_m}, \mathbf{h}_{S_1}, \ldots, \mathbf{h}_{S_n}\}$. These node features undergo $N$ standard attention modules, encoded as the final node embeddings $\{\mathbf{h}_{D_1}^{(N)}, \ldots, \mathbf{h}_{D_m}^{(N)}, \mathbf{h}_{S_1}^{(N)}, \ldots, \mathbf{h}_{S_n}^{(N)}\}$ for downstream decoding.

**MDLRAM-decoder:** With the encoded node embeddings, the decoder operates iteratively to construct feasible solution routes in form of vertices' permutation as an MDP. Each decoding step necessitates a ***context embedding*** $\mathbf{h}_c^t$ depicting current state $s_t$, and a ***mask*** finalizing point selection domain through filtering out the current infeasible points, both updated step-wisely.

*(i) Context embedding*: We design the context embedding $\mathbf{h}_c^t$ to depict current state, concatenating four elements: $\mathbf{h}_c^t = W^c[\mathbf{h}_a \| \mathbf{h}_{(t)} \| \mathbf{h}_{D(t)} \| Q_t] + \mathbf{b}^c$, where $\mathbf{h}_a = \frac{1}{m+n}(\sum_{k=1}^m \mathbf{h}_{D_k}^{(N)} + \sum_{e=1}^n \mathbf{h}_{S_e}^{(N)})$ is the global information; $\mathbf{h}_{(t)}$ is the node embedding of the point where current vehicle is situated, while $Q_t$ is the remained load on current vehicle. Notably, $\mathbf{h}_{D(t)}$ is the node embedding of the depot which current route belongs to.

*(ii) Mask mechanism*: In each decoding step, guided by the context embedding $\mathbf{h}_c^t$, the decoder produces the corresponding probabilities for all the feasible points within the selection domain, while infeasible points—determined by vehicle remained load and tasks completion state—are masked. To efficiently handle batch processing of problem instances, we employ a boolean mask tailored for the LRP scenario, allowing for batch-wise manipulation on selection domains, avoiding repeated operation for each individual instance. (See step-wise update pseudo code in Appendix A.3)

Upon finalizing the boolean mask for current decoding step, the context embedding is applied to conduct Multi-head Attention (MHA) with the node embeddings filtered by the mask. This yields an intermediate context embedding $\hat{\mathbf{h}}_c^t$ incorporating the glimpse information on each feasible point. Then, $\hat{\mathbf{h}}_c^t$ participates in Single-head Attention (SHA) with the filtered node embeddings, yielding the corresponding probabilities for all the feasible points in its selection domain, where a feasible point, as an action $a_t$, can be selected with an associated $p_t$. This decoding process is delineated as:

$$\hat{\mathbf{h}}_c^t = \text{FF}(\text{MHA}(\mathbf{h}_c^t, \text{mask}\{\mathbf{h}_{D_1}^{(N)}, \ldots, \mathbf{h}_{D_m}^{(N)}, \mathbf{h}_{S_1}^{(N)}, \ldots, \mathbf{h}_{S_n}^{(N)}\}))$$

$$a_t = \text{argmax}(\text{softmax}[\frac{1}{\sqrt{\dim}} \cdot \text{FF}_{(\text{query})}(\hat{\mathbf{h}}_c^t) \cdot \text{FF}_{(\text{key})}(\text{mask}\{\mathbf{h}_{D_1}^{(N)}, \ldots, \mathbf{h}_{D_m}^{(N)}, \mathbf{h}_{S_1}^{(N)}, \ldots, \mathbf{h}_{S_n}^{(N)}\})^T])$$

$$(2)$$

## 2.2 DUAL-MODE DEPOT GENERATION: DGM

As depicted in the purple block of Fig. 1, the DGM is designed to only take in the customer requests data and generate the depots in two distinct modes based on preference: exact depot locations or a multivariate Gaussian distribution for flexible depot sampling.

**DGM-Configuration:** The configuration for depot generation is also defined on an undirected graph $G = (V, E)$, where the $V = \{\mathbf{v}_{S_1}, \ldots, \mathbf{v}_{S_n}\}$ only including customer requests. A solution set incorporating $m$ depots is pending to be generated. During depot generation, the distances among generated depots are expected to be within the range $[l_{\min}, l_{\max}]$, which means the depots being excessively close or distant with each other will both incur violation penalty.

**DGM-Objective Function:** As the main task of depot generation, the depots with desired properties are expected to be generated. According to the problem configuration for DGM, for the solution set of generated depots, denoted as $\mathcal{D}$, its loss can be defined as Eq. (3), where $L_{\text{MDLR}}$ is the route length derived by MDLRAM based on the DGM generated depots, $\lambda, \varepsilon$ are coefficients for penalty of the depots being too distant or close with each other:

$$L_{\text{Gen}}(\mathcal{D}) = L_{\text{MDLR}} + \sum_{i=1}^{m} \sum_{j=i}^{m} [\lambda \cdot \max(d_{ij} - l_{\max}, 0) + \varepsilon \cdot \max(l_{\min} - d_{ij}, 0)] \tag{3}$$

DGM aims to minimize the expectation of this loss associated with generated depot set, defined as $\mathbb{E}[L_{\text{Gen}}(\mathcal{D})]$, where $L_{\text{Gen}}(\mathcal{D})$ is formed as Eq. (3).

**DGM-encoder:** The DGM solely processes the customer requests, each characterized by a vector $\mathbf{g}_{S_e} = [x_{S_e}, y_{S_e}, q_e]^T$ concatenating location and demands. Following the similar encoding process with MDLRAM, these requests are encoded as node embeddings $\{\tilde{\mathbf{h}}_{S_1}^{(N)}, \ldots, \tilde{\mathbf{h}}_{S_n}^{(N)}\}$, based on which a global embedding is finalized as: $\mathbf{h}_{serve} = \frac{1}{n} \sum_{i=1}^{n} \tilde{\mathbf{h}}_{S_i}^{(N)}$ for downstream depot generation.

**DGM-generator in Multivariate Gaussian distribution mode:** In this mode, the DGM aims to generate a multivariate Gaussian distribution where depots can be flexibly sampled. Since $m$ depots are pending to be identified, the generated multivariate Gaussian distribution should exhibit $2m$ dimensions, with each pair of dimensions denoting the coordinates $(x_{D_k}, y_{D_k})$ for depot $D_k$, where $k \in \{1, \ldots, m\}$. To achieve this, we define this multivariate Gaussian distribution, pending to be generated, as: $\mathbf{X}_{depot} \sim \mathcal{N}(\boldsymbol{\mu}, \Sigma)$, where any randomly sampled $2m$-dimensional vector $\mathbf{X}_{depot} = (X_1, X_2, \ldots, X_{2m})^T$ represents the coordinates for a depot set including $m$ depots.

To generate such distribution, two essential components are: the mean vector $\boldsymbol{\mu} \in \mathbb{R}^{2m}$ and covariance matrix $\Sigma \in \mathbb{R}^{2m \times 2m}$. Hence, the output of DGM should be a vector $\mathbf{h}_{depot}$ as below, where the first $2m$ dimensions represent the mean vector, followed by the second $2m$ dimensions denote corresponding variance of each coordinate, with the remaining $C_{2m}^2$ dimensions as the covariance of any two coordinates. Therefore, the $\mathbf{h}_{depot} \in \mathbb{R}^{2m+2m+C_{2m}^2}$ is arranged as Eq. (4).

$$\mathbf{h}_{depot} = (\overbrace{h_1, \ldots, h_{2m}}^{mean}, \overbrace{\ldots, h_{4m}}^{variance}, \overbrace{\ldots, h_{4m+C_{2m}^2}}^{covariance})^T \tag{4}$$

To output this $\mathbf{h}_{depot}$ for constructing the $2m$-dimensional Gaussian distribution, we employ a layer module featuring two fully connected layers to process the global embedding $\mathbf{h}_{serve}$ derived by encoder, expressed as: $\mathbf{h}_{depot} = \tanh(\text{FF}_{\text{multiG}}(\tanh(\text{FF}(\mathbf{h}_{serve}))))$.

When utilizing the $\mathbf{h}_{depot}$ to construct the multivariate Gaussian distribution, it is critical to ensure that the variances remain positive. Hence, before constructing, we process the second $2m$ dimensions' elements as below: $\vec{var} = 1 + \text{elu}((h_{2m+1}, \ldots, h_{4m})^T)$. Besides, when sampling the depot set $\mathbf{X}_{depot}$ which records the depot coordinates, to adhere to the configuration, $\mathbf{X}_{depot}$ should be mapped within unit square $[0, 1] \times [0, 1]$ to standardize the depot set: $\mathcal{D}_{\text{multiG}} = \text{sigmoid}(\mathbf{X}_{depot})$.

**DGM-generator in Exact position mode:** In this mode, DGM aims to directly generate the exact positions for a set of depots based on the global embedding $\mathbf{h}_{serve}$ derived by encoder. To this end, we retain $m$ as the depot number, then the DGM's output should be a vector $\mathbf{h}_{depot} \in \mathbb{R}^{2m}$ in which every two dimensions represent the coordinates $(x_{D_k}, y_{D_k})$ for a depot $D_k$.

$$\mathbf{h}_{depot} = (\overbrace{h_1, h_2}^{D_1}, \ldots, \overbrace{h_{2m-1}, h_{2m}}^{D_m})^T \tag{5}$$

To output this $\mathbf{h}_{depot}$, a layer module, encompassing two fully connected layers, is employed: $\mathbf{h}_{depot} = \text{FF}_{\text{exactP}}(\tanh(\text{FF}(\mathbf{h}_{serve})))$. Also, to satisfy the configuration, we map the $\mathbf{h}_{depot}$ within the unit square $[0,1] \times [0,1]$ to standardize the depot set: $\mathcal{D}_{\text{exactP}} = \text{sigmoid}(\mathbf{h}_{depot})$.

## 3 TRAINING STRATEGY

**Step I: Pre-training of MDLRAM:** MDLRAM concurrently processes a batch of problem instances randomly sampled from the configuration, thereby generating a batch of corresponding MDPs as their respective feasible solutions. Each MDP involves a permutation of actions, denoted as $\text{MDP}(\mathbf{A}) = \{a_1, a_2, \dots\}$. Because each action $a_t$ is associated with a probability $p_t$ for selecting the corresponding point, the entire MDP's probability is manifested as: $p_{\boldsymbol{\theta}_{\text{I}}}(\mathbf{A}) = \prod_t p_t = \prod_t p(s_{t+1}|s_t, a_t)$, which is parameterized by $\boldsymbol{\theta}_{\text{I}}$, denoting the MDLRAM's parameters that require training. Based on a batch of such MDPs, each associated with a probability $p_{\boldsymbol{\theta}_{\text{I}}}(\mathbf{A})$ and a cost $L_{\text{Sel}}(\mathbf{A})$ in Eq. (1), the MDLRAM is trained by REINFORCE gradient estimator with greedy roll-out baseline Kool et al. (2019) to minimize the expectation of cost, as depicted in Eq. (6), where the baseline $\bar{\mathcal{B}}$ is established through a parallel network mirroring the structure of MDLRAM, persistently preserving the best parameters attained and remaining fixed. (See Appendix A.4 for pseudo code and details.)

$$\nabla\mathcal{L}(\boldsymbol{\theta}_{\text{I}}) = \mathbb{E}_{p_{\boldsymbol{\theta}_{\text{I}}}(\mathbf{A})}[(L_{\text{Sel}}(\mathbf{A}) - \bar{\mathcal{B}})\nabla \log p_{\boldsymbol{\theta}_{\text{I}}}(\mathbf{A})] \tag{6}$$

**Step II: Dual-mode training of DGM:** As depicted in Fig. 2, DGM can be trained in two modes, with the pre-trained MDLRAM serving as a fixed sub-solver. **For the record**, the $\boldsymbol{\theta}_{\text{II}}$, denoting DGM's parameters that requires training, is appended as footnote only to those variables which are parameterized by DGM.

*(i) Multivariate Gaussian distribution mode:* In this mode, as depicted in left side of Fig. 2, the DGM takes in a main-batch (Batchsize: $B_{\text{main}}$) of randomly sampled graphs: $\{G_b | b = 1, 2, ..., B_{\text{main}}\}$, each with a group of customer requests, and then outputs a main-batch of corresponding multivariate Gaussian distributions: $\{\mathcal{N}_b(\boldsymbol{\mu}, \Sigma) | b = 1, 2, ..., B_{\text{main}}\}$. Thus, training DGM involves ensuring that the depots sampled from these distributions yield favorable expectations for the cost $L_{\text{Gen}}(\mathcal{D})$.

To achieve this, from each distribution $\mathcal{N}_b(\boldsymbol{\mu}, \Sigma)$ within the main-batch, we sample a sub-batch (Batchsize: $B_{\text{sub}}$) sets of depots. Each depot set is represented as $\mathcal{D}_{\text{multiG}}$, associated with their probabilities $p_{\boldsymbol{\theta}_{\text{II}}}(\mathcal{D}_{\text{multiG}})$ and costs $L_{\text{Gen}}(\mathcal{D}_{\text{multiG}})$ in Eq. (3). In this way, a main-batch ($B_{\text{main}}$) of cost expectations, each corresponding to a multivariate Gaussian distribution, can be derived. This entire process is shown in left part of Fig. 2. We employ following optimizer to train the DGM in this distribution mode:

$$\nabla\mathcal{L}(\boldsymbol{\theta}_{\text{II}}) = \frac{1}{B_{\text{main}}} \sum_{b=1}^{B_{\text{main}}} \mathbb{E}_{p_{\boldsymbol{\theta}_{\text{II}}}(\mathcal{D}_{\text{multiG}})}^{(b)}[L_{\text{Gen}}(\mathcal{D}_{\text{multiG}}, G_b) \cdot \nabla \log p_{\boldsymbol{\theta}_{\text{II}}}(\mathcal{D}_{\text{multiG}})] \tag{7}$$

*(ii) Exact position mode:* In this mode, as depicted in right side of Fig. 2, the DGM still ingests a main-batch ($B_{\text{main}}$) of randomly sampled graphs: $\{G_b | b = 1, 2, ..., B_{\text{main}}\}$, each with a group of customer requests, but directly generates the corresponding sets of depots: $\{\mathcal{D}_{\text{exactP}}^{(b)} | b = 1, 2, ..., B_{\text{main}}\}$. For each set of depots $\mathcal{D}_{\text{exactP}}$, the cost $L_{\text{Gen}}(\mathcal{D}_{\text{exactP}})$ can be derived by Eq. (3) whose first part is obtained by pre-trained MDLRAM. Below optimizer guides the DGM's training in exact mode:

$$\nabla\mathcal{L}(\boldsymbol{\theta}_{\text{II}}) = \frac{1}{B_{\text{main}}} \sum_{b=1}^{B_{\text{main}}} \nabla L_{\text{Gen}}((\mathcal{D}_{\text{exactP}}^{(b)})_{\boldsymbol{\theta}_{\text{II}}}, G_b) \tag{8}$$

Figure 2: DGM's dual-mode training.

***It is crucial to differentiate that***, for different modes, the DGM's parameters $\boldsymbol{\theta}_{\text{II}}$ are tracked in different variables. In multivariate Gaussian distribution mode, what has been parameterized by $\boldsymbol{\theta}_{\text{II}}$ is the probability for each sampled $\mathcal{D}_{\text{multiG}}$, whereas in exact position mode, what has been parameterized by $\boldsymbol{\theta}_{\text{II}}$ is the $\mathcal{D}_{\text{exactP}}$. Therefore, the gradients in these two modes are respectively backpropagated to parameters $\boldsymbol{\theta}_{\text{II}}$ through $p_{\boldsymbol{\theta}_{\text{II}}}(\mathcal{D}_{\text{multiG}})$ and $(\mathcal{D}_{\text{exactP}})_{\boldsymbol{\theta}_{\text{II}}}$.

# 4 EXPERIMENTAL RESULTS AND DISCUSSION

**Training Setup:** To ensure comparability with prior methods, we establish the training dataset following the setup outlined in prior routing studies Kool et al. (2019). As for the depot-related setting, we adopt the data format prevalent in real-world LRP benchmark datasets which are conventionally employed in pertinent studies Belenguer et al. (2011); Prins et al. (2006). Every single *problem instance* in the training dataset is defined on a unit square $[0, 1] \times [0, 1]$, where the customers' requests are uniformly scattered, with their corresponding demands uniformly sampled from $[0, 10]$.

The *problem instances* for MDLRAM's pre-training are from three problem scales: $n = 20, 50, 100$ customers, respectively coupled with $m = 3, 6, 9$ depot candidates. ***Corresponding to each scale:*** (1) The vehicle's maximum capacity $Q$ is selected as $30, 40, 50$ respectively; (2) The vehicle's setup cost $U$ is set as 0.3; (3) The depot's desired maximum supply $M_k$ is uniformly selected from $[50, 80], [80, 120], [120, 170]$; (4) The depot's opening cost $O_k$ is uniformly selected from $[2, 5]$, $[2, 5], [12, 19]$; The coefficients in objective function Eq. (1) are defined as $\alpha = 1, \beta = 1, \delta = 2$; For each scale, we train MDLRAM on one A40 GPU for 100 epochs with 1,280,000 *problem instances* generated on the fly as training dataset, which can be split into 2,500 batches with batchsize of 512 (256 for scale 100 due to device memory limitation).

As for DGM's *problem instance*, only including customer requests, we also consider three problem scales: $n = 20, 50, 100$ customers. The expected distance among the generated depots ranges within $[0.2, 0.7]$. The coefficients in objective function Eq. (3) are specified as $\lambda = 10, \varepsilon = 10$. Correspondingly, for each scale, we train DGM on one A40 GPU for 100 epochs. Within each epoch, 2,500 main-batches of *problem instances* are generated on the fly as training dataset and iteratively fed into DGM. ***In multivariate Gaussian distribution mode***, the main-batch size $B_{\text{main}}$ is set as 32 (16 for scale 100), and the sub-batch size $B_{\text{sub}}$ for sampling in each distribution is selected as $128, 64, 32$ for scale $20, 50, 100$ respectively. ***In exact position mode***, where no sampling is performed, we set main-batch size as 512 (256 for scale 100).

## 4.1 RESULTS ANALYSIS FOR CRITIC MODEL - MDLRAM

We first assess the efficacy of the pre-trained MDLRAM. As critic model, it is expected to instantly provide optimized LRP solution for serving customers based on generated depots, in batch-wise manner. Therefore, we test it on both ***Synthetic Dataset*** to highlight the instant batch-wise solving ability, and ***Real-world Benchmark Dataset*** to evaluate its generalization performance by comparing with SOTA results, which, so far, are all achieved by specifically designed heuristic methods.

**(i) Testing on Synthetic dataset:** For each problem scale, the synthetic testing dataset include 10,000 problem instances randomly sampled from the same configuration in training process, capable of being divided into batches to facilitate batch-wise testing. Given the lack of existing DRL method specifically designed for standard LRP scenario setup, we compare results from various enhanced classic heuristic methods, which are commonly applied to solve routing problems, even though heuristic methods are not suitable for batch-wise usage considering its solving manner and unstable inference time. The parameters are tuned to align LRP setup to report the best performance.

Table 1: Batch-wise Testing Performance of MDLRAM on Synthetic Dataset. ("Ttl.C.: total cost in Eq.(1)"; "Len.: total length"; "Dpt.C. (Nb.): depot opening cost (opened depot number)"; "Veh.C. (Nb.): vehicle setup cost (used vehicle numbers)"; "Dpt.P.: penalty for exceeding depot desired maximum supply")

| | $n$ | $m$ | Mtd. | Ttl.C. | Len. | Dpt.C. | (Nb.) | Veh.C. | (Nb.) | Dpt.P. | Inf.T. |
|---|---|---|---|---|---|---|---|---|---|---|---|
| **Scale 20** | 20 | 3 | MDLR(S) | **13.17\*** | 5.61 | 6.26 | (2.14) | 1.23 | (4.08) | 0.07 | 0.16s |
| | 20 | 3 | MDLR(G) | 13.81 | 5.86 | 6.45 | (2.15) | 1.25 | (4.17) | 0.24 | 0.10s |
| | 20 | 3 | ALNS | 13.23 | 6.56 | 5.40 | (2.01) | 1.16 | (3.88) | 0.10 | 4.12s |
| | 20 | 3 | GA | 13.72 | 6.87 | 5.61 | (2.01) | 1.17 | (3.90) | 0.07 | 6.63s |
| | 20 | 3 | TS | 16.26 | 9.13 | 5.79 | (2.00) | 1.16 | (3.88) | 0.18 | 0.16s |
| **Scale 50** | 50 | 6 | MDLR(S) | **20.85\*** | 8.65 | 10.13 | (3.52) | 2.05 | (6.82) | 0.03 | 0.43s |
| | 50 | 6 | MDLR(G) | 22.05 | 9.20 | 10.66 | (3.55) | 2.09 | (6.97) | 0.10 | 0.31s |
| | 50 | 6 | ALNS | 25.10 | 14.71 | 8.33 | (3.03) | 2.05 | (6.84) | 0.01 | 47.35s |
| | 50 | 6 | GA | 29.65 | 18.53 | 9.03 | (3.12) | 2.09 | (6.98) | 0 | 12.69s |
| | 50 | 6 | TS | 33.26 | 22.26 | 8.91 | (3.16) | 2.10 | (6.99) | 0 | 1.57s |
| **Scale 100** | 100 | 9 | MDLR(S) | **86.88\*** | 14.79 | 68.77 | (5.00) | 3.32 | (11.06) | 0 | 1.04s |
| | 100 | 9 | MDLR(G) | 94.94 | 16.29 | 75.00 | (5.00) | 3.37 | (11.23) | 0.28 | 0.55s |
| | 100 | 9 | ALNS | 89.75 | 29.58 | 56.82 | (3.97) | 3.21 | (10.69) | 0.14 | 412.83s |
| | 100 | 9 | GA | 108.24 | 43.79 | 61.03 | (4.14) | 3.28 | (10.92) | 0.14 | 19.44s |
| | 100 | 9 | TS | 104.39 | 41.80 | 59.31 | (4.08) | 3.27 | (10.89) | 0.01 | 16.44s |

As presented in Table 1, the testing on synthetic dataset provides a detailed breakdown of cost and inference times for various problem scales, with customer numbers indexed as $n$ ranging from 20 to 100, and depot numbers marked as $m$ varying from 3 to 9. For each scale, all methods involved in

comparison report their average objective value on the 10,000 synthetic problem instances. Specifically, for MDLRAM, the results are reported in two testing strategies, decided by its decoding process: **(a) Greedy test:** when generating the solution routes for each problem instance, the action selected in each decoding step is the point with the highest probability, thereby deriving one greedy solution; **(b) Sampling test:** For each instance, MDLRAM simultaneously generates 1,280 random solutions by stochastically select action in each decoding step. Then, the one with the lowest cost is chosen as the solution.

From Table 1, MDLRAM's sampling test consistently achieves the lowest total cost across all scales, outperforming other methods. Meanwhile, its greedy way yields smaller total cost than other methods on larger scale $n = 50, 100$, being slightly outperformed on scale $n = 20$. The error bars for the greedy test results are $\pm 0.11$, $\pm 0.11$, and $\pm 0.35$ for scales 20, 50, and 100, respectively. Regarding each individual objective, all methods exhibit similar vehicle usage, but MDLRAM tends to distribute this usage among more depots, resulting in an increase on depot opening cost compared to heuristic methods. This indicates that DRL method's extensive searching ability enables exploration of a broader range of circumstances, leading to better solutions. As for inference time, with the increase of scale, MDLRAM shows steady performance, basically within 1s timeframe, whereas heuristic methods demonstrate significant increase on time consumption.

**(ii) Testing on Real-world dataset:** To further assess the MDLRAM's generalization performance on real-world problem instances with diverse node distribution compared to the synthetic problem instances used during training, we conduct individual comparison on instances from four real-world datasets (Prodhon Prins et al. (2006), Acka Akca et al. (2009), Tuzun Tuzun & Burke (1999), Barreto Barreto et al. (2007)) which include their best-known solutions (BKS), and the SOTA results derived by existing methods. *Notably,* as a critic model, MDLRAM stands out for its rapidity to plan high-quality solutions in batches, which lays the foundation for depot-generating tasks completed by DGM. Therefore, when testing on real-world dataset, our aim is not to establish new SOTA results, instead, we

Table 2: MDLRAM's Performance on cases from Real-world Dataset ("*" represents an optimal solution; "Ttl.C.": Total Cost; "Inf.T.": Inference Time).

| Case name | $n$ | $m$ | BKS | GRASP Prins et al. (2006) | | | MDLRAM (ours) | | |
|---|---|---|---|---|---|---|---|---|---|
| | | | | Ttl C. | Gap | Inf.T. | Ttl C. | Gap | Inf.T. |
| P111122 | 100 | 20 | 14492 | 15269.0 | 5.36% | 40.7s | 15554.5 | 7.33% | 0.75s |
| P111222 | 100 | 20 | 14323 | 14822.9 | 3.49% | 36.2s | 15154.43 | 5.80% | 0.74s |
| P111112 | 100 | 10 | 14676.8 | 15252.5 | 3.92% | 32.4s | 15516.6 | 5.72% | 0.64s |
| P113122 | 100 | 20 | 12463 | 12729.4 | 2.14% | 36.0s | 13081.49 | 4.96% | 0.72s |
| P111212 | 100 | 10 | 13948 | 14235.4 | 2.06% | 27.6s | 14529.15 | 4.17% | 0.58s |
| 50-5-1a | 50 | 5 | 90111 | 90632 | 0.57% | 1.8s | 95072 | 5.51% | 0.25s |
| 50-5-2b | 50 | 5 | 67340 | 68042 | 1.04% | 2.5s | 70941 | 5.35% | 0.23s |
| 50-5-3b | 50 | 5 | 61830 | 61890 | 0.10% | 2.0s | 66258 | 7.16% | 0.23s |
| G67-21-5 | 21 | 5 | 424.9* | 429.6 | 1.1% | 0.2s | 425.66 | 0.18% | 0.12s |
| 20-5-1a | 20 | 5 | 54793* | 55021 | 0.42% | 0.2s | 57005 | 4.04% | 0.11s |
| 20-5-2a | 20 | 5 | 48908* | 48908 | 0.00% | 0.1s | 50029 | 2.29% | 0.12s |
| 20-5-2b | 20 | 5 | 37542* | 37542 | 0.00% | 0.2s | 38893 | 3.60% | 0.11s |

| Case name | $n$ | $m$ | BKS | HBP Akca et al. (2009) | | | MDLRAM (ours) | | |
|---|---|---|---|---|---|---|---|---|---|
| | | | | Ttl C. | Gap | Inf.T. | Ttl C. | Gap | Inf.T. |
| P183-12-2 | 12 | 2 | 204* | 204.0 | 0.00% | 0.2s | 204.00 | 0.00% | 0.07s |
| P183-55-15 | 55 | 15 | 1112.06 | 1121.8 | 0.88% | 10800s | 1151.91 | 3.58% | 0.38s |
| P183-85-7 | 85 | 7 | 1622.5 | 1668.2 | 2.82% | 10813.8s | 1676.89 | 3.35% | 0.46s |

| Case name | $n$ | $m$ | BKS | B&C Belenguer et al. (2011) | | | MDLRAM (ours) | | |
|---|---|---|---|---|---|---|---|---|---|
| | | | | Ttl C. | Gap | Inf.T. | Ttl C. | Gap | Inf.T. |
| 30-5a-1 | 30 | 5 | 819.52* | 819.60 | 0.00% | 50.22s | 849.33 | 3.64% | 0.143s |
| 30-5a-2 | 30 | 5 | 821.50* | 823.50 | 0.00% | 53.89s | 884.29 | 7.64% | 0.144s |
| 40-5a-1 | 40 | 5 | 928.10* | 928.20 | 0.00% | 305.25s | 988.80 | 6.54% | 0.189s |
| 40-5b-1 | 40 | 5 | 1052.04* | 1052.07 | 0.00% | 3694.45s | 1107.54 | 5.28% | 0.195s |

demonstrate how MDLRAM consumes significantly less inference time than existing method to derive high-quality solutions comparable to BKS, thereby ensuring the efficacy for depot generation.

As detailed in Table 2, these instances diverse significantly, with customer amount $n$ ranging from 12 to 100 and depot candidate amount $m$ from 2 to 20. For each case, we juxtapose: the BKS, the results achieved by the specifically designed SOTA method reported in literatures, and results obtained through our MDLRAM. Across all the cases, our approach can plan solution routes comparable to those of traditional method but with notably reduced inference time. This efficiency becomes increasingly pronounced as the problem scale enlarges, demonstrating MDLRAM's capability to maintain solution quality while significantly reducing time consumption.

## 4.2 RESULTS ANALYSIS FOR DGM

As for the DGM, based on a same group of customer requests devoid of predefined depot candidates, it is expected to generate a depot set $\mathcal{D}$ which can lead to lower total cost than the randomly attempted depots. Thus, we arrange an experiment to evaluate which of the following three strategies can identify the best depot set for a same group of customer requests: (i) Generating the depot set in DGM-Exact mode; (ii) Generating the depot set in DGM-Gaussian mode; (iii) Randomly attempting the depot set in batches. The quality of the depot set is judged by $L_{\text{Gen}}(\mathcal{D})$ in Eq. (3). ***For each scale*** $n \in \{20, 50, 100\}$, we randomly sample 8,000 problem instances as testing dataset. Each instance only includes a group of customer requests. The results are reported in two ways, respectively

evaluating the *average-level* and *best-level* of the solution depot set generated by each method: **(i) Average test:** For each problem instance, these methods respectively generate 512 solution sets of depots to serve the same group of customers accordingly. The mean of these 512 cost values is reported as the corresponding result for this problem instance. Then, the average of these 8,000 mean costs are obtained as final result. **(ii) Sampling test:** Similarly, for each problem instance, these methods respectively generate 512 solution sets of depots to serve the same group of customers, but only the lowest cost is reported as the result for that problem instance. The final result is derived as the average of these 8,000 lowest cost values. ***Notably, DGM's Exact mode directly generate deterministic solution depot set for given problem instance, thereby yielding identical outcomes for its both testing ways.***

All three methods undergo testing on same dataset, and their results are respectively decomposed and compared in Table 3 with two testing ways: **(1) Average test** aims at comparing the ***average level*** of the solution depot set that each method can generate. Observations reveal that both DGM's two modes can identify superior solution depot set than randomly attempting, while the Exact mode exhibits better performance over the Gaussian mode; **(2) Sampling test** further compares the ***best level*** of solution depot set that each method can achieve. The results demonstrate that, when compared with "randomly attempting" on the same problem instance, DGM's Gaussian mode can find better solution depot set within the same sampling timeframe. As for the DGM's Exact mode, its specific solution set con-

Table 3: Comparison of solution depot set generated by different strategies. ("Ttl.C.: total cost in Eq.(3)"; "Len.: total length"; "Ex.P. or Ls.P.: penalty for exceeding Upper bound or Lower bound of the distance among the generated depots";)

| | $n$ | Method | Ttl C. | Len. | Ex. P. | Ls. P. | (Dpt C. | Veh C. | Dpt P.) |
|---|---|---|---|---|---|---|---|---|---|
| **Average** | | Rdm. | 7.671 | 5.857 | 1.121 | 0.693 | (6.448 | 0.246 | 1.249) |
| | 20 | DGM-G. | **5.724** | **5.370** | 0.354 | 0.000 | (6.427 | 0.234 | 1.252) |
| | | DGM-E. | **5.096*** | **5.089*** | 0.005 | 0.002 | (6.449 | 0.242 | 1.251) |
| | | Rdm. | 18.261 | 9.192 | 5.600 | 3.469 | (10.650 | 0.101 | 2.090) |
| | 50 | DGM-G. | **10.921** | **8.660** | 0.339 | 1.922 | (10.649 | 0.090 | 2.092) |
| | | DGM-E. | **8.531*** | **8.496*** | 0.018 | 0.017 | (10.637 | 0.082 | 2.083) |
| | | Rdm. | 38.040 | 16.281 | 13.434 | 8.325 | (74.956 | 0.281 | 3.371) |
| | 100 | DGM-G. | **23.305** | **14.616** | 0.896 | 7.793 | (74.933 | 0.271 | 3.372) |
| | | DGM-E. | **15.394*** | **13.690*** | 0.026 | 1.678 | (74.940 | 0.274 | 3.385) |
| **Sample** | | Rdm. | **5.045*** | **5.021*** | 0.014 | 0.010 | (6.465 | 0.270 | 1.247) |
| | 20 | DGM-G. | 5.137 | 5.115 | 0.022 | 0.000 | (6.428 | 0.249 | 1.249) |
| | | DGM-E. | 5.096 | 5.089 | 0.005 | 0.002 | (6.449 | 0.242 | 1.251) |
| | | Rdm. | 10.684 | 8.570 | 0.610 | 1.504 | (10.679 | 0.083 | 2.086) |
| | 50 | DGM-G. | **8.769** | **8.259*** | 0.135 | 0.375 | (10.676 | 0.085 | 2.086) |
| | | DGM-E. | **8.531*** | **8.496** | 0.018 | 0.017 | (10.637 | 0.082 | 2.083) |
| | | Rdm. | 24.056 | 14.910 | 2.491 | 6.655 | (74.928 | 0.281 | 3.369) |
| | 100 | DGM-G. | **19.538** | **13.828** | 0.667 | 5.043 | (74.942 | 0.250 | 3.367) |
| | | DGM-E. | **15.394*** | **13.690*** | 0.026 | 1.678 | (74.940 | 0.274 | 3.385) |

sistently outperform those of the Gaussian mode, whereas the Gaussian mode can offer more flexibility with its sampling ability. Only one exception is observed in small scale ($n = 20$), where "randomly attempting" achieves better solution set without the aid of DGM, which may not be replicable at larger scales. ***It's worth noting that,*** the superior performance achieved by the generated depot set $\mathcal{D}$ not just simply reflects on the total cost $L_{\text{Gen}}(\mathcal{D})$ which is the sum of route length and the violation of distance range among depots, but also respectively reflected on each individual cost items. This reflects DGM's ability on both identifying good depot positions and satisfying the location requirements, instead of only focusing on minimizing the violation of distance range among depots, while neglecting the route length, to achieve "superior performance".

**Visualize Depots Distribution:** DGM's Gaussian mode reveals correlations between depot coordinates through learnable covariances. Visualization of these distributions shows that for smaller problem scales $m = 3, n = 20$, the 6-D normal distribution tends to present as distinct 2-D normal distributions. However, as problem scales grow, the relationships between depot coordinates become more complex, instead of simply presenting as several discrete 2-D normal distributions, implying that, at larger scales, random sampling would require significant computational effort to cover optimal depots. Full details and visualizations can be found in Appendix B.2.

## 5 CONCLUSION AND FUTURE WORK

In this study, we propose a generative DRL framework for depot generation without predefined candidates. Based on customer requests data, the DGM proactively generates depots, while the MDLRAM efficiently plans routes from these generated depots, demonstrating flexibility and cost reductions, especially in scenarios requiring quick depot establishment and flexible adjustments. This modular framework can be adapted to various LRP variants and further optimized for inter-depot cost balancing (see Appendix B.3 for extended results). For more detailed discussions on the framework's limitations and future work, such as incorporating additional depot constraints and generating depots adaptive to multiple routing tasks, please refer to the Appendix B.4.

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

APPENDIX

# A ADDITIONAL DETAILS ABOUT METHODOLOGY

## A.1 LRP CONFIGURATION

**Assumptions:** Following the established assumptions Belenguer et al. (2011): (1) Each customer's demand must be served by a delivery from exactly one depot and load transfers at intermediate locations are not allowed; (2) Each customer must be served exactly once by one vehicle, i.e., splitting order is not allowed; (3) No limits on the number of vehicles utilized, but the vehicle cost should be minimized as part of the objective.

**Constraints:** The constraints in LRP includes three aspects. (1) Customer Demand: The vehicle's remaining capacity must suffice to cover its next target customer's demand during service; (2) Vehicle Capacity: The cumulative demands delivered in a single vehicle route cannot surpass the vehicle's maximum capacity; (3) Depot Supply: The aggregate demands dispatched from a specific depot is expected not to exceed its desired maximum supply.

***Remark 1:*** *The first two items are hard constraints determining solution feasibility, whereas the last item is a soft constraint manifesting as a penalty in the objective function.*

## A.2 MDP FORMULATION

Here, we propose the formulation of feasible LRP solution routes in form of MDP, which is an entire permutation of the vertices in the graph. As depicted in Fig. 3, the routes corresponding to the same depot have the identical start and end point, facilitating their aggregation into an entire permutation by jointing their identical depot. Consequently, by linking together these permutations from all depots, a feasible solution can be finally formulated as an MDP.

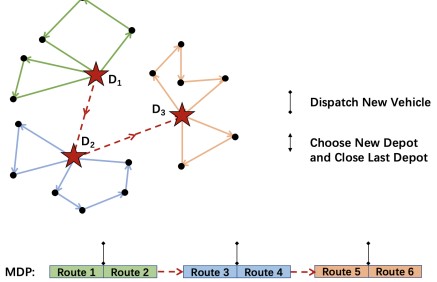

***Remark 2:*** *The MDP is a necessary mathematical formulation used to construct the feasible solution routes when engaging DRL method. Once the solution is derived in MDP form, it will be reverted to a set of routes for simultaneous execution by multiple vehicles.*

Figure 3: The feasible LRP solution in this example consists of 6 single routes, which are simultaneously carried out by multiple vehicles. The routes in same color belongs to a same depot. By linking them together, the feasible solution is formulated in points permutation, as an MDP.

We define this MDP with a tuple $(\mathbf{S}, \mathbf{A}, \mathbf{P}, \mathbf{R}, \gamma)$, where, in each decision step $t$, the current iteration is represented by a tuple $(s_t, a_t, p_t, r_t, \gamma_t)$.

(a) $\mathbf{S}$ : is a set of states, wherein each state corresponds to a tuple $(G, D_t, \mathbf{v}_t, Q_t)$, where $G$ denotes entire static graph information; $D_t$ indicates the depot which current route belongs to; $\mathbf{v}_t$ signifies current customer in decision step $t$; $Q_t$ records remaining capacity on current vehicle; This tuple is updated at each decision step within MDP.

(b) $\mathbf{A}$ : is a set of actions, wherein each action $a_t$ is the next point that current vehicle plans to serve. In this problem configuration, to ensure that the MDP represents a feasible solution, actions should be selected from feasible points whose demands can be satisfied by current vehicle's remaining capacity. Upon selecting the $a_t$, the state tuple should be updated accordingly:

$$Q_{t+1} = \begin{cases} Q_t - q_e & \text{if } a_t \in \{\mathbf{v}_{S_e} | e = 1, 2, \ldots, n\}, \\ Q & \text{if } a_t \in \{\mathbf{v}_{D_k} | k = 1, 2, \ldots, m\}, \end{cases} \quad (9)$$

$a_t \in \{\mathbf{v}_{S_e} | e = 1, 2, \ldots, n\}$ indicates that current vehicle is scheduled to visit an unserved customer. Then, the remaining capacity $Q_t$ should be updated according to Eq. (9), wherein $q_e$ represents the demand associated with the customer selected by action $a_t$. Meanwhile, $a_t \in \{\mathbf{v}_{D_k} | k =$

$1, 2, \ldots, m\}$ indicates that current vehicle chooses to return to its departure depot, or start planning for a new depot. Then, a new vehicle's route will commence from this depot, thereby the capacity $Q_t$ is refreshed to full state.

(c) $\mathbf{P}$ : is a set of probabilities, wherein each element $p_t$ represents the probability transiting from state $s_t$ to $s_{t+1}$ by taking action $a_t$, and $p_t$ can be expressed as: $p_t = p(s_{t+1}|s_t, a_t)$

(d) $\mathbf{R}$ : is a set of costs, wherein each element $r_t$ denotes the cost incurred by taking action $a_t$ in step $t$. The $r_t$ can be expressed as follows, where $d_{ij}$ denotes the length between $\mathbf{v}_i$ in step $t$ and $\mathbf{v}_j$ in step $t + 1$:

$$r_t = \begin{cases} 0 & \text{if } \mathbf{v}_i, \mathbf{v}_j \in \{\mathbf{v}_{D_k}|k = 1, 2, \ldots, m\}, \\ d_{ij} & \text{otherwise,} \end{cases} \tag{10}$$

As is shown in Eq. (1), apart from this step-wisely accumulated transit distance, other costs used to depict the overall performance of the solution routes, which are not accumulated step-wisely, are added into the total cost after an entire MDP is generated. These additional overall costs include: *(i) the opening cost for used depots; (ii) the setup cost for dispatched vehicles; (iii) penalty of exceeding depot desired maximum supply*.

(e) $\gamma \in [0, 1]$ : the discount factor for cost in each step. Here, we presume no discount applies to the costs, i.e., $\gamma = 1$

### A.3 MULTI-DEPOT MASK MECHANISM

In each decoding step, guided by the context embedding $\mathbf{h}_c^t$, the decoder produce the corresponding probabilities for all the feasible points within the selection domain. This selection domain should exclude all the points that current vehicle cannot visit in next step, which is subject to vehicle capacity and current state in MDP. Because the model processes problem instances in batches, simultaneous updates to their respective selection domains at each decoding iteration is necessary.

We identify four key scenarios to categorize the selection domain of each instance at any given step, based on the vehicle's location (depot or customer) and the completion status of delivery tasks. Specifically, these four potential patterns are summarized as follows:

- (i) When current vehicle is at a depot and all the customers' delivery tasks are finished: it can only stay at current depot.

- (ii) When current vehicle is at a depot but not all the customers' delivery tasks are finished: it can choose from the vertices set including all the unserved customers and unplanned depots but excluding current depot.

- (iii) When current vehicle is at a customer and all the customers' delivery tasks are finished: this represents the current customer is the last delivery task, implying that the only selection is the vehicle's departure depot.

- (iv) When current vehicle is at a customer but not all the customers' delivery tasks are finished: it can choose from the vertices set including all the unserved customers and its departure depot.

Based on these four patterns, the selection domain is updated before each decoding iteration.

As discussed, the model operates in batch-wise manner, necessitating simultaneous updating each instance's selection domain at each decoding iteration. **The challenge is**, in each decoding step, the selection domain of each problem instance within one batch can be very different. Thus, an efficient boolean mask matrix specific to the LRP scenario is devised for batch-wise manipulation on selection domain, avoiding repeated operation on individual problem instance.

The Algorithm 1 specifies our mask mechanism specifically tailored for LRP scenario. which includes manipulations on the selection domain of customers and depots. Firstly, by masking the customers which have been served or cannot be satisfied with remaining capacity, the selection domain of customers can be simply derived. Crucially, for the depot selection domain, we notice that among the four patterns above: three patterns (i, iii, and iv) include only the departure depot, whereas one pattern (ii) excludes the departure depot. Thus, at each decoding step for a batch of

instances, we initially mask all the depots unanimously and only reveal their departure depot of current routes. Then, we identify the problem instances belonging to pattern-ii in this batch, mask the departure depots and reveal the unplanned depots. All the manipulations operate in batches to avoid repeated operation on individual problem instance.

---

**Algorithm 1** Mask Mechanism for batch-wise manipulation on selection domain for a batch of problem instances

---

**Input**: A batch of problem instances with Batch Size $B$

1: **Init** Record $= [\sigma_{ij}] \in \mathbb{R}^{B \times (m+n)}$ where $\sigma_{ij} \in \{0,1\}$ representing, in problem instance $i$, whether the vertex $j$ is visited ($\sigma_{ij} = 0$) or unvisited ($\sigma_{ij} = 1$)
2: **Init** ID $\in \mathbb{R}^B$ current situated vertices for all instances
3: **Init** DP $\in \mathbb{R}^B$ current departure depots for all instances
4: **for** each decoding step $t = 1, 2, ...$ **do**
5:     $\{\varphi_i\} \leftarrow$ Batch No. for the problem instances where not all the tasks are finished
6:     $\{\varphi_j\} \leftarrow$ Batch No. for the problem instances where all the tasks are finished
7:     $\sigma_{ij} \leftarrow 0$ according to the $\text{ID}_t$
8:     $(\text{Mask}_0)_{ij} \leftarrow True \; if \; \sigma_{ij} = 0, (\text{Mask}_0)_{ij} \leftarrow False \; if \; \sigma_{ij} = 1$
9:     $(\text{Mask}_1)_{ij} \leftarrow True \; if \; (Q_t)_i < (q_e)_j, (\text{Mask}_0)_{ij} \leftarrow False \; if \; (Q_t)_i > (q_e)_j$
10:    $\text{Mask} \leftarrow \text{Mask}_0 + \text{Mask}_1$
11:    $(\text{Mask})_{ij} \leftarrow True \; for \; all \; j \in \{0, 1, ..., m-1\}$
12:    $(\text{Mask})_{ij} \leftarrow False$ according to the $\text{DP}_t$
13:    $\{\varphi_k\} \leftarrow$ Batch No. for the problem instances where current vertex is one of the depots
14:    $\{\varphi_e\} \leftarrow \{\varphi_i\} \cap \{\varphi_k\}$ Batch No. for the problem instances where current vertex is one of the depots and not all tasks are finished
15:    $(\text{Mask})_{ij} \leftarrow False$ where $i \in \{\varphi_e\}$ and $j \in \{0, 1, ..., m-1\}$
16:    $(\text{Mask})_{ij} \leftarrow True$ where $i \in \{\varphi_e\}$ and $\text{DP}_{\varphi_e} \in \{0, 1, ..., m-1\}$
17:    $(\text{Mask})_{ij} \leftarrow True$ where $j \in \{0, 1, ..., m-1\}$ and $\sigma_{ij} = 0$
18: **end for**
19: **Return** Mask

---

## A.4 MDLRAM's pre-training & DGM's dual-mode training

---

**Algorithm 2** Pre-training for MDLRAM

---

**Input**: $M$ batches of problem instances with Batch Size $B$

1: **for** each epoch $ep = 1, 2, ..., 100$ **do**
2:   **for** each batch $bt = 1, 2, ..., M$ **do**
3:     $\{G_b | b = 1, 2, ..., B\} \leftarrow$ A Batch of Cases
4:     $\{A_b^{\theta_\mathrm{I}} | b = 1, 2, ..., B\} \leftarrow \text{MDLRAM}_{\theta_\mathrm{I}}(\{G_b\})$
5:     $\{A_b^{\theta_\mathrm{I}^*} | b = 1, 2, ..., B\} \leftarrow \text{MDLRAM}_{\theta_\mathrm{I}^*}(\{G_b\})$
6:     $\nabla \mathcal{L}(\boldsymbol{\theta}_\mathrm{I}) \leftarrow \frac{1}{B} \sum_{b=1}^{B} [(L(A_b^{\theta_\mathrm{I}}) - L(A_b^{\theta_\mathrm{I}^*})) \nabla \log p_{\boldsymbol{\theta}_\mathrm{I}}(A_b^{\theta_\mathrm{I}})]$
7:     **if** One Side Paired T-test $(A_b^{\theta_\mathrm{I}}, A_b^{\theta_\mathrm{I}^*}) < 0.05$ **then**
8:       $\theta_\mathrm{I}^* \leftarrow \theta_\mathrm{I}$
9:     **end if**
10:   **end for**
11: **end for**

---

The baseline $\bar{\mathcal{B}}$ in Algorithm 2 is established through a parallel network mirroring the structure of MDLRAM, persistently preserving the best parameters attained and remaining fixed. Parameters' update solely occurs if a superior evaluation outcome is derived by MDLRAM, enabling baseline network's adoption of these improved parameters from MDLRAM. The actions in MDPs produced by MDLRAM is selected with probabilistic sampling in each decoding step, whereas that of baseline network is greedily selected based on the maximum possibility.

---

**Algorithm 3** Dual-mode training for DGM, coupled with pretrained MDLRAM functioning as a critic model

---

**Input**: Batches of problem instances with Batch Size $B_{\text{main}}$

---

1: **if** in Multivariate Gaussian Distribution mode **then**
2:    **for** each epoch $ep = 1, 2, ..., 100$ **do**
3:       **for** each batch $bt = 1, 2, ..., M$ **do**
4:          $\{G_b | b = 1, 2, ..., B_{\text{main}}\} \leftarrow$ A Main-Batch of graphs with customers Info
5:          $\{\mathcal{N}_b^{\theta_{\text{II}}} | b = 1, 2, ..., B_{\text{main}}\} \leftarrow \text{DGM}_{\theta_{\text{II}}}(\{G_b\})$
6:          **for** each graph $b = 1, 2, ..., B_{\text{main}}$ **do**
7:             $\{\mathcal{D}_{\text{multiG}}^{(b')} | b' = 1, 2, ..., B_{\text{sub}}\} \leftarrow$ A Sub-Batch of sampled depot sets
8:             $\nabla L_{DGM}(\mathcal{N}_b) \leftarrow \mathbb{E}_{p_{\theta_{\text{II}}}(\mathcal{D}_{\text{multiG}})}^{(b)}[\text{MDLRAM}(\mathcal{D}_{\text{multiG}}^{(b')}, G_b)$
9:             $\cdot \nabla \log p_{\theta_{\text{II}}}(\mathcal{D}_{\text{multiG}}^{(b')})]$
10:          **end for**
11:          $\nabla \mathcal{L}(\theta_{\text{II}}) \leftarrow \frac{1}{B_{\text{main}}} \sum_{b=1}^{B_{\text{main}}} \nabla L_{DGM}(\mathcal{N}_b)$
12:       **end for**
13:    **end for**
14: **else if** in Exact Position mode **then**
15:    **for** each epoch $ep = 1, 2, ..., 100$ **do**
16:       **for** each batch $bt = 1, 2, ..., M$ **do**
17:          $\{G_b | b = 1, 2, ..., B_{\text{main}}\} \leftarrow$ A Main-Batch of graphs with customers Info
18:          $\{\mathcal{D}_{\text{exactP}}^{(b)} | b = 1, 2, ..., B_{\text{main}}\} \leftarrow \text{DGM}_{\theta_{\text{II}}}(\{G_j\})$
19:          $\nabla \mathcal{L}(\theta_{\text{II}}) \leftarrow \frac{1}{B_{\text{main}}} \sum_{b=1}^{B_{\text{main}}} \nabla \text{MDLRAM}((\mathcal{D}_{\text{exactP}}^{(b)})_{\theta_{\text{II}}}, G_b)$
20:       **end for**
21:    **end for**
22: **end if**

---

## B  EXTENDED DETAILS ABOUT EXPERIMENTAL RESULTS

### B.1  HYPERPARAMETERS DETAILS

For MDLRAM, we train it for 100 epochs with training problem instances generated on the fly, which can be split into 2500 batches with batchsize of 512 (256 for scale 100 due to device memory limitation). Within each epoch, by going through the training dataset, MDLRAM will be updated 2500 iterations. After every 100 iterations, the MDLRAM will be assessed on an evaluation dataset to check whether improved performance is attained. The evaluation dataset consists of 20 batches of problem instances, with the same batch size of 512(256).

For DGM, we also train it for 100 epochs. In each epoch, 2500 main-batches of problem instances are iteratively fed into DGM. **In multivariate Gaussian distribution mode**, the main-batch size $B_{\text{main}}$ is set as 32 (16 for scale 100), and the sub-batch size $B_{\text{sub}}$ for sampling in each distribution is selected as 128, 64, 32 for scale 20, 50, 100 respectively. During training, after every 100 iterations' updating, the DGM will be evaluated on an evaluation dataset to check if a better performance is derived. The evaluation dataset is set as 20 main-batches of problem instances, maintaining the same batch size $B_{\text{main}}$ and $B_{\text{sub}}$. **In exact position mode**, where no sampling is performed, we set main-batch size as 512 (256 for scale 100). Likewise, after every 100 iterations' updating, an evaluation process is conduct on 20 main-batches of problem instances with corresponding batch size of 512 (128) to check if DGM achieves a better performance.

As for the hyperparameters in model architecture across the entire framework, the encoding process employs $N = 3$ attention modules with 8-head MHA sublayers, featuring an embedding size of 128. All the training sessions are finished on one single A40 GPU.

Parameters for heuristic methods in Table 1: **(a)** Adaptive Large Neighborhood Search (ALNS): *Destroy (random percentage $0.1 \sim 0.4$, worst nodes $5 \sim 10$); Repair (random, greedy, regret with 5 nodes); Rewards ($r_1 = 30, r_2 = 20, r_3 = 10, r_4 = -10$); Operators weight decay rate: $0.4$; Threshold decay rate: $0.9$;* **(b)** Genetic Algorithm (GA): *Population size: 100; Mutation probability:*

*0.2; Crossover probability: 0.6;* **(c)** Tabu Search (TS): *Action Strategy (1-node swap, 2-node swap, Reverse 4 nodes); Tabu step:* 30;

## B.2 VISUALIZE DEPOTS DISTRIBUTION:

DGM's distribution mode is trained to understand correlations between coordinates of various depots, manifested as their learnable covariances. To visualize the distribution generated in the Gaussian mode of DGM and observe how this multivariate Gaussian distribution is represented in a 2-D graph, we depict the generated multivariate Gaussian distribution for problem instances from all three scales. A notable pattern is revealed as below:

In the problem scale of $m = 3, n = 20$, the 6-D normal distribution tends to present as three separate 2-D normal distributions, as depicted in Fig. 4. However, as the problem scales increase, such as the 12-D ($m = 6, n = 50$) or 18-D ($m = 9, n = 100$) normal distributions, they do not tend to present as several discrete 2-D normal distributions.

***This trend indicates that***, in large-scale scenario, the covariance between coordinates from different depots exhibit a more complex relationship, which further implies that simply relying on randomly sampling depots in pursuit of covering optimal depots would require an expansive search and substantial computational effort.

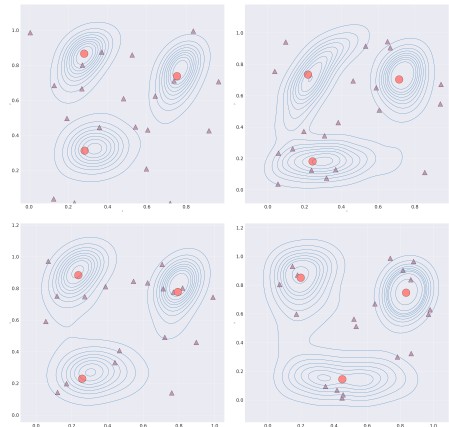

Figure 4: Visualization of Multivariate Gaussian Distribution outputted by DGM based on customer requests (Gray): Predicted Depot Distribution (Blue), and Optimal Depots Identified (Red).

## B.3 MDLRAM'S ABILITY ON BALANCING ROUTE LENGTH AMONG DEPOTS

With MDLRAM's structure, fine-tuning the model to align with diverse additional requirements associated to the multiple depots in LRP scenario is flexible through designing specialized cost functions. Here, we examine the route balancing challenge among various depots.

If the objective is to maintain the route length $l_k(\mathbf{A})$ associated with each depot $D_k$ ($k \in \{1, 2, \ldots, m\}$) in a specific proportional relationship, namely $l_1(\mathbf{A}) : l_2(\mathbf{A}) : \ldots : l_m(\mathbf{A}) = \rho_1 : \rho_2 : \ldots : \rho_m$, while simultaneously minimizing the overall cost $L_{\text{Sel}}(\mathbf{A})$, it can be achieved by augmenting the cost function $L_{\text{Sel}}(\mathbf{A})$ in Eq. (1) with a balance penalty as follows:

$$\tilde{L}_{\text{Sel}}(\mathbf{A}) = L_{\text{Sel}}(\mathbf{A}) + \sum_{k=1}^{m} \sum_{k'=k}^{m} \left| l_k(\mathbf{A}) - \frac{\rho_k}{\rho_{k'}} l_{k'}(\mathbf{A}) \right| \tag{11}$$

To evaluate the adaptability of MDLRAM in addressing LRP with additional requirements on adjusting inter-depot cost distribution, we fine-tune the MDLRAM, which has been pre-trained with original objective $L_{\text{Sel}}(\mathbf{A})$ in Eq. (1), with this new balance-oriented objective $\tilde{L}_{\text{Sel}}(\mathbf{A})$ in Eq. (11) on the same training dataset. In this context, our specific goal is to ensure that the lengths belonging to each depot are approximately equal (i.e., $\rho_k = 1$). Notably, $\rho_k$ can be adjusted based on specific proportion requirements.

To illustrate the effectiveness of balance-oriented fine-tuning, we select random cases from each scale for direct comparison of route length belonging to each depot, generated by MDLRAM under different objectives. In Table 4, it can be observed that, for each case, the balance penalty of solution routes found by MDLRAM under balance-oriented objective Eq. (11) is conspicuously smaller than that of original objective Eq. (1), only incurring a slight wave on the total length as an acceptable trade-off for incorporating the additional item in the balance-oriented objective function. This can also be directly reflected by the balanced route length distribution across depots in 5th column of Table 4.

Table 4: Comparison of Each Depot's Route Length, respectively planned by Original MDLRAM and the Fine-tuned Version. ("Obj.": Objective Function; "Ori.Obj.": Original Objective Function in Eq. (1); "Bln.Obj.": Balance-oriented Objective Function in Eq. (11); "Bln.Pen.": penalty for measuring the balancing performance of route length among depots; "Dpt.Nb.": opened depot number out of total available depots).

| | Case | Obj. | Bln. Pen. | (Dpt Nb.) | Saperate Depot Len. | Total Len. |
|---|---|---|---|---|---|---|
| scale 20 | case1 | Ori obj. | 0.758 | 2/3 | 3.487-2.729 | 6.216 |
| | | **Bln obj.** | **0.008** | 2/3 | 2.781-2.772 | 5.554 |
| | case2 | Ori obj. | 0.929 | 2/3 | 3.439-2.511 | 5.951 |
| | | **Bln obj.** | **0.007** | 2/3 | 3.022-3.016 | 6.038 |
| | case3 | Ori obj. | 0.926 | 2/3 | 3.608-2.682 | 6.290 |
| | | **Bln obj.** | **0.022** | 2/3 | 3.123-3.102 | 6.225 |
| | case4 | Ori obj. | 0.693 | 2/3 | 2.853-2.159 | 5.012 |
| | | **Bln obj.** | **0.0002** | 2/3 | 2.518-2.518 | 5.036 |
| scale 50 | case1 | Ori obj. | 3.131 | 4/6 | 2.158-2.536-2.155-3.073 | 9.922 |
| | | **Bln obj.** | **0.129** | 4/6 | 2.492-2.530-2.521-2.507 | 10.052 |
| | case2 | Ori obj. | 3.738 | 4/6 | 2.150-3.154-2.947-2.220 | 10.471 |
| | | **Bln obj.** | **0.283** | 4/6 | 2.449-2.434-2.473-2.383 | 9.739 |
| | case3 | Ori obj. | 2.016 | 3/6 | 2.981-2.579-3.586 | 9.146 |
| | | **Bln obj.** | **0.067** | 3/6 | 3.085-3.091-3.058 | 9.234 |
| | case4 | Ori obj. | 2.416 | 4/6 | 1.808-2.596-1.918-1.969 | 8.292 |
| | | **Bln obj.** | **0.176** | 4/6 | 2.190-2.186-2.163-2.220 | 8.759 |
| scale 100 | case1 | Ori obj. | 3.444 | 5/9 | 2.728-3.132-2.496-3.092-2.642 | 14.091 |
| | | **Bln obj.** | **0.916** | 5/9 | 2.736-2.742-2.829-2.842-2.915 | 14.063 |
| | case2 | Ori obj. | 2.495 | 5/9 | 3.008-3.344-3.063-3.487-3.353 | 16.256 |
| | | **Bln obj.** | **0.373** | 5/9 | 3.045-3.015-2.987-2.987-2.967 | 15.001 |
| | case3 | Ori obj. | 5.310 | 5/9 | 3.743-2.622-2.985-3.335-2.922 | 15.606 |
| | | **Bln obj.** | **1.641** | 5/9 | 3.043-3.099-3.056-3.249-3.358 | 15.808 |
| | case4 | Ori obj. | 8.711 | 5/9 | 3.273-3.398-4.455-2.599-2.754 | 16.479 |
| | | **Bln obj.** | **1.896** | 5/9 | 3.492-3.465-3.404-3.709-3.755 | 17.825 |

## B.4 FURTHER DISCUSSION

In this study, we extend the exploration of the LRP by addressing a real-world challenge: the generation of depots when no predefined candidates are presented. For this purpose, a generative DRL framework comprising two models is proposed. Specifically, the DGM, based on customer requests data, enables proactive depot generation with dual operational modes flexibly- the exact mode ensures precision when necessary, while the Gaussian mode introduces sampling variability, enhancing the model's generalization and robustness to diverse customer distributions. Meanwhile, the MDL-RAM subsequently facilitates rapid planning of LRP routes from the generated depots for serving the customers, minimizing both depot-related and route-related costs. Our framework represents a transition from traditional depot selection to proactive depot generation, showcasing cost reductions and enhanced adaptability in real-world scenarios like disaster relief, which necessitates quick depot establishment and flexible depot adjustment.

The framework's detachability offers flexible extension for its application. The DGM's depot-generating ability can be fine-tuned to adapt different LRP variants by jointing with other downstream models, making DGM a versatile tool in real-world logistics. Meanwhile, the end-to-end nature of MDLRAM enable its flexible usage on addressing LRP variants with requirements of adjusting inter-depot cost distribution, which has been detailed in Appendix B.3.

Based on the framework design details and the application scenario description, we spot following limitatioins and arranging a research landscape for future works.

**Limitation:** While the MDLRAM model has the ability to select a flexible number of depots from the generated depot set when planning routes for vehicle from the generated depot set, the number

of depots generated by the DGM is currently set fixed during training. Incorporating an adaptive mechanism within the DGM to dynamically determine the optimal number of depots based on customer demands and logistical factors could further enhance the framework's flexibility and efficiency. Achieving this adaptive depot generation may require a more conjugated and interactive integration between the DGM and the MDLRAM's route planning process.

**Future work:** Future research will focus on expanding DGM's applicability by incorporating a wider range of depot constraints to reflect more real-world scenarios accurately. For example, in this study, we consider the distance between depots should adhere to a specific range requirements, preventing the depots from being too close or too distant with each other. Additional constraints on depots can be emphasized on the forbidden area within the map, such as ensuring the depots are not situated in specific regions or must be placed within designated zones.

Additionally, leveraging the framework's modular design to adapt to various routing tasks presents an exciting avenue for exploration. This includes generating depots which can generally achieve satisfying performance across multiple concurrent routing tasks, which would further extend the framework's utility in complex and dynamic real-world logistics environments.

