# OpenReview forum: "Beyond Predefined Depots: A Dual-Mode Generative DRL Framework for Proactive Depot Generation in Location-Routing Problem"
_ICLR.cc/2025/Conference — Submitted to ICLR 2025_

### Official Review · Reviewer_8oe9 · 2024-10-28

**Soundness:** 2
**Presentation:** 2
**Contribution:** 2
**Rating:** 3
**Confidence:** 5

**Summary:**

The paper proposes a generative deep reinforcement learning (DRL) method for solving an integrated location-and-routing problem in transportation and logistics. It tests the results using synthetic and real-world datasets.

**Strengths:**

The authors try to apply methods from the machine learning and artificial intelligence (ML/AI) domain to provide efficient and intuitive results for classical NP-hard location-and-routing problem in Operations Research. The paper aims to overcome the computational challenges of traditional optimization models and approaches through model-free approaches.

**Weaknesses:**

While the paper aims to bridge the ML/AI and traditional optimization via solving a specific classical problem in transportation, the design of the DRL-based method is mainly heuristic driven. The paper is more close to an academic exercise, rather than an in-depth study. It involves a depot generative model that uses demand information to sample depot locations, and a critic model that scores the generated depot locations to further output location-routing solutions. The motivation behind such a two-module design is missing and by having the two separated procedures (i.e., location and then routes), the optimality of the solutions based on the Deep Reinforcement Learning for the integrated location-routing problem does not seem to be guaranteed anymore. The paper also lacks theoretical analysis about the optimality bound guarantee nor the efficiency of the algorithm. The training strategies are also ad hoc and do not have a firm theoretical ground.

At a high level, while location-routing is a class of classical transportation problems, the version considered in this paper is not considered state of the art, and the literature now has been focusing on optimizing location solutions and controlling dynamic routes (in an online fashion) under uncertain demand and route information. Therefore, how the proposed methods add to the existing knowledge is not clear to me.

**Questions:**

1.	How the proposed DRL based method compares with classical discrete optimization solutions for location-routing problem?  Any theoretical optimality guarantee that one can prove (even under special problem structures or networks)?
2.	The numerical studies are comparing with other three approaches from papers using either heuristics or branch and cut published in years 2006, 2009, 2011, for different sizes of networks, but these are not recent work in the related literature and cannot represent the state-of-the-art. Any benchmarks with methods published more recently?
3.	Can the DRL handle very large-scale instances? What is the solution quality and computational efficiency tradeoff?
4.	Why deep reinforcement learning? The motivation and connection of DRL with location-routing are not sufficient nor clear.

---

> ### Author Response · Authors · 2024-11-17
> **Response to Reviewer 8oe9 – Part 1**
>
> Thank you for taking the time to review our paper. We appreciate your insights and would like to address the concerns you've raised point-by-point.
>
> __1. Motivation and connection of DRL with location-routing:__ Thanks for the opportunity to clarify the motivation behind using DRL in the context of LRP, we will delineate this from following three aspects:
>
> - __(1) Why using DRL in routing problem:__
>
>   - __(i) Cost-Effectiveness Balance:__ Different methods—exact algorithms, heuristics, and DRL—have their own advantages and limitations, which is why we consider the cost-effectiveness balance when comparing methods for routing problem. For different routing variants with diverse complication level, this balance shifts, making new methods come into view.
>
>   - __(ii) Factors involved in Cost-Effectiveness balance:__  Traditionally, considerations of cost-effectiveness include computational cost, inference time, and routing performance. This was also how the heuristic method initially comes along to complement the intensive computation of exact method. Now, beyond these factors, as problem scales and complexity continue to increase, additional factors like batch-wise computing and generality (i.e., reduced effort required to adapt existing methods to new scenarios) also become important in the cost-effectiveness balance. This is where DRL can offer significant advantages, help alleviating two dilemmas for traditional methods:
>     - __(a) Repetitive Search Processes:__ Traditional methods typically require initiating a new search procedure for each individual case, even with minor alterations, leading to a case-by-case solving approach. This is not desirable in dynamic scenario with large problem scales.
>     - __(b) Adaptability to Variants:__ Expanding a specific VRP variant algorithm to accommodate other specialized variants with different task configurations often lacks adaptability in traditional methods.
>
>   - __(iii) Key Advantages of DRL:__ We certainly appreciate the strengths of exact and heuristic methods. However, when certain more important advantages are difficult to achieve with traditional methods, DRL becomes a valuable alternative to attain a more desirable cost-effectiveness balance.The key advantages of DRL present in following points:
>     - __(a) Batch-wise Computing:__ DRL models can process multiple instances simultaneously, improving computational efficiency and making them suitable for large-scale problems and dynamic scenario.
>     - __(b) Learning Desirable Solution Patterns through Sequential Decision-Making Framework:__ The Markov Decision Process (MDP) framework enables DRL models to capture problem configurations and sequentially generate solutions, allowing more flexible adaptation to various routing variants.
>     - __(c) Stable Performance After Training:__ Once trained, the model can solve new cases with same task configurations with stable performance, achieving a good balance between inference time and solution quality.
>
>   - __(iv) Advancements of DRL in recent routing research:__ Building on these advantages, the advancements in learning-based methods have shown promise in addressing routing problems, both in learn-to-construct [1][2], learn-to-generalize [3][4], learn-to-improve [5][6], and learn-to-decompose [7][8]. It also shows good performance and efficiency on various routing problem variants, such as heterogeneous routing [9], drone-aided routing [10], etc.
>
> [Reference List]
>
> [1] Kool et al. Attention, learn to solve routing problems! ICLR, 2019.
>
> [2] Xin et al. Multi-decoder attention model with embedding glimpse for solving vehicle routing problems. AAAI, 2021.
>
> [3] Lin et al. Cross-problem learning for solving vehicle routing problems. IJCAI, 2024.
>
> [4] Zhou et al. Multi-task vehicle routing solver with mixture-of-experts. ICML, 2024.
>
> [5] Lu et al. A learning-based iterative method for solving vehicle routing problems. ICLR, 2020.
>
> [6] Ma et al. Learning to iteratively solve routing problems with dual-aspect collaborative transformer. NeurIPS, 2021.
>
> [7] Ye et al. Glop: Learning global partition and local construction for solving large-scale routing problems in real-time. AAAI, 2024.
>
> [8] Hou et al. Generalize learned heuristics to solve large-scale vehicle routing problems in real-time. ICLR, 2023
>
> [9] Li et al. Deep reinforcement learning for solving the heterogeneous capacitated vehicle routing problem. IEEE T-cyber, 2021.
>
> [10] Bogyrbayeva et al. A deep reinforcement learning approach for solving the traveling salesman problem with drone. TR-C, 2023

---

> > ### Comment · Reviewer_8oe9 · 2024-11-21
> > **How to evaluate DRL over optimization methods without really comparing with the state of the art?**
> >
> > The response does not address my concerns regarding the contribution of the work. The citations on location-routing problem considered in this work and methods they benchmark with were mostly published in lower-tier journals in the field of Operations Research and Management Science, which has a long tradition of studying vehicle routing, facility location, and combined location routing problems. I am surprised to see that no papers from, e.g., Transportation Science (TS), Operations Research (OR), Transportation Research Part B (TRB), etc., where cited, not in the paper, nor in this new response. This reflected the authors' lack of knowledge in this domain, and if arguing about "cost effectiveness" "batch-wise computing" "solutions patterns" "stable performance" etc., as the hypothetical advantages of DRL, how the authors know that they were not offered in the state-of-the-art optimization literature? There were papers on tackling multistage stochastic/robust VRP, LRP, etc., under the uncertainty of travel time, demand, etc., with or without time window constraints; papers of offline or online algorithms for these problems; papers aiming to final optimal policies or approximations with bounded solution optimality; and so on. For sequential decision making, there were papers using Approximate Dynamic Programming, to address the issue of dimensionality, etc. I list some examples below and some of them are papers already published in the OR literature 15+ years ago, not to mention the more recent ones focusing on understanding both solution optimality and also computational efficiency published in TS, OR, TRB, etc. In fact, the exact approaches can solve much larger size instances than the ones considered in this paper. Without benchmark with these approaches (either theoretically or numerically, beyond just generic branch-and-cut tested in this paper), it lacks rigor to state that DRL outperforms exact methods because they are more stable, or cost effective, etc.
> >
> > Drexl, M., & Schneider, M. (2015). A survey of variants and extensions of the location-routing problem. European journal of operational research, 241(2), 283-308.
> >
> > Desaulniers, G., Errico, F., Irnich, S., & Schneider, M. (2016). Exact algorithms for electric vehicle-routing problems with time windows. Operations Research, 64(6), 1388-1405.
> >
> > Ulmer, M. W. (2017). Approximate dynamic programming for dynamic vehicle routing (Vol. 61). Berlin/Heidelberg, Germany: Springer International Publishing.
> >
> > Novoa, C., & Storer, R. (2009). An approximate dynamic programming approach for the vehicle routing problem with stochastic demands. European journal of operational research, 196(2), 509-515.
> >
> > Powell, W. B. (2007). Approximate Dynamic Programming: Solving the curses of dimensionality (Vol. 703). John Wiley & Sons.
> >
> > Powell, W. B., Simao, H. P., & Bouzaiene-Ayari, B. (2012). Approximate dynamic programming in transportation and logistics: a unified framework. EURO Journal on Transportation and Logistics, 1(3), 237-284.

---

> > > ### Author Response · Authors · 2024-11-22
> > > **Discussion with Review 8oe9 – Part 1**
> > >
> > > Thank you for engaging with our response and providing further feedback. We respectfully find that some of your comments reflect misunderstandings or mischaracterizations of our work. We would like to address these concerns with clarity and evidence.
> > >
> > > __1. Validity of Citations and Recognized Benchmarks in LRP Research:__
> > >
> > > - __(1)__ You remarked that our benchmarks and results are from "low-tier" venues. However, the Location-Routing Problem benchmarks and methodologies we cite—particularly from the European Journal of Operational Research (EJOR) and Computers & Operations Research—are recognized and used within the LRP research community. These citations are not arbitrarily chosen; they represent real-world benchmark dataset and are foundational to comparative studies in this domain, for instance:
> > >
> > >   - Barreto, S., Ferreira, C., Paixão, J., & Santos, B. S. (2007). "Using clustering analysis in a capacitated location-routing problem." European Journal of Operational Research, 179(3), 968–977.
> > >   - Belenguer, J.-M., Benavent, E., Prins, C., Prodhon, C., & Wolfler Calvo, R. (2011). "A branch-and-cut method for the capacitated location-routing problem." Computers & Operations Research, 38(6), 931–941.
> > >
> > > - __(2)__ You also kindly listed Drexl and Schneider's (2015) survey, which is also from European Journal of Operational Research (EJOR) – one venue of our benchmark resource. This survey includes the works and benchmark we have cited, as well as the works which you concern as a publication on low-tier venues. This suggests alignment between our work and the foundational literature in the LRP field, underscoring the validity of our chosen references.
> > >
> > >   - Prins, C., Prodhon, C., & Wolfler Calvo, R. (2006). "Solving the capacitated location-routing problem by a GRASP complemented by a learning process and a path relinking." 4OR, 4(3), 221–238.
> > >   - Akca, Z., Berger, R. T., & Ralphs, T. K. (2009). "A branch-and-price algorithm for combined location and routing problems under capacity restrictions." In Operations Research and Cyber-Infrastructure (pp. 309–330). Springer.
> > >
> > > __2. Focus on Problem Scope:__
> > >
> > > - We acknowledge that the journals you listed (TS, OR, TRB) are indeed prestigious, but none of the papers that you listed from these journals appear to address the LRP without predefined depot candidates or provide widely accepted benchmarks for this specific problem. The absence of such works in our citations does not indicate a lack of domain knowledge but rather reflects the specific scope of our work.
> > >
> > > - Our work specifically targets the LRP without predefined depot candidates, focusing on proactive depot generation by learning desirable depot distribution patterns over customer groups. This is a less-explored area that necessitates referencing the most relevant and recognized literature, which we have done.
> > >
> > > - We are open to incorporating additional references if you would point us to specific papers from these journals that align with our LRP scope and include recognized benchmarks or datasets relevant to our problem. We would be eager to include them in our validation.
> > >
> > > __3. Claims of "Hypothetical Advantages" of DRL and the “general” Branch-and-Cut Method:__
> > >
> > > - __(1) Claims of "Hypothetical Advantages" of DRL :__ You characterized the advantages of DRL presented in our paper—such as cost-effectiveness, batch-wise computation, and stable performance—as hypothetical. However, these advantages are substantiated by the extensive experimental results presented in Table 1&2 of our paper, using comparison on real-world datasets to demonstrate its ability to approximate optimal routing costs while achieving significant reductions in inference time, while also incorporating additional synthetic experiments to evaluate batch-wise processing capabilities, consistency, stability, and reliability of the solutions.
> > >
> > > - __(2) Branch-and-Cut (B&C) Comparisons in Table 2:__ You remarked that the B&C method used in Table-2 as general. However, this B&C method used in our benchmarks is not a generic implementation. Specifically, it is a published and specialized method from Computers & Operations Research tailored to the LRP context. This method is recognized within the LRP community and is also included in the EJOR survey that you cited. We carefully selected this approach for its relevance and recognition in LRP research, ensuring a fair and meaningful comparison with our DRL-based method.
> > >
> > >   - Belenguer, J.-M., Benavent, E., Prins, C., Prodhon, C., & Wolfler Calvo, R. (2011). "A branch-and-cut method for the capacitated location-routing problem." Computers & Operations Research, 38(6), 931–941.

---

> > > ### Author Response · Authors · 2024-11-22
> > > **Discussion with Review 8oe9 – Part 1- follow up**
> > >
> > > __4. Acknowledgment of Variants and Methodologies:__
> > >
> > > We acknowledge that, in route planning domain, numerous variants integrating various constraints and objectives from customers, fleet owners, and regulators, and are addressed by various types of methods. Each variant corresponds to different application scenarios with specific requirements and operational targets. We believe that potential applications have their own significance and consideration on method selection, even if they have not yet been identified or extensively studied yet.
> > > While we appreciate the diversity of variants and approaches in this domain, we would like to clarify the scope of our work as follows:
> > >
> > > - __Problem Configuration:__ Our paper specifically addresses the LRP without predefined depot candidates, focusing on proactive depot generation through learning desirable depot distribution patterns. Many of the references you provided pertain to the Vehicle Routing Problem (VRP) or stochastic problem variants, which, while important, are outside the primary scope of our research.
> > >
> > > - __Methodology:__ While we appreciate the contributions of approximate dynamic programming and exact optimization methods to related problems, the references you provided do not address the specific constraints and objectives we are investigating. We also value your suggestions on various potential exact solution ideas to provide insights, but these are neither archived as a recognized reference nor within our DRL-based routing research scope.

---

> ### Author Response · Authors · 2024-11-17
> **Response to Reviewer 8oe9 – Part 2**
>
> __[Following Part 1]__
>
> - __(2) Significance of LRP without Predefined Depot Candidates:__ While some current research focuses on dynamic routing and uncertainty, the LRP without predefined depot locations remains a less-explored area with substantial practical relevance. By investigating this problem in literatures (Page3-Line127) where we identify the practical application scenario and the unsolved constraints, we aim to broaden the scope of LRP research and provide solutions for scenarios that are not adequately addressed by existing methods. Specifically:
>
>   - __(i) Practical Applications Requiring Rapid Depot Establishment:__ As detailed in the introduction part, there are critical real-world situations where the assumption of predefined depot locations is not feasible. Examples include emergency medical rescue and disaster relief operations, where infrastructure may be compromised or non-existent. In such cases, there is an urgent need to determine or adjust depot locations quickly and efficiently to facilitate effective response efforts.
>
>   - __(ii) Constraint of Computational Intensity:__ Regarding this extended LRP scenario without predefined depot candidates, the existing methods predominantly employing heuristic strategies for attempting new depots across a planar area. In this situation, if simply expanding the search of the depot candidates across the map and aimlessly attempting new points, then undesired increasing on problem scale will be incurred, thereby leading to excessive time consumption and expensive computation. Therefore, devising a method to proactively generate high-quality depots, satisfying the depot location constraints for LRP scenario without predefined depot candidates, is well-motivated. Our approach addresses this need by efficiently generating depot locations without exhaustive searching.
>
> - __(3) How DRL method contribute to advance the LRP without predefined depots candidates:__
>
>   - __(i) Proactive Generation through DRL's Generative Capability, instead of attempting:__ Existing methods predominantly employ heuristic strategies that attempt depot placements across the map. This approach can lead to an undesirable increase in problem scale, resulting in excessive time consumption and computational expense due to the need to evaluate numerous potential depot locations individually. Our approach leverages the generative capabilities of DRL to proactively generate high-quality depot locations. By doing so, we avoid repetitive and aimless searching, reducing computational complexity and improving efficiency. The DGM generates depot locations based on learned representations of customer demand, streamlining the process and focusing on promising areas.
>
>   - __(ii) Understanding Problem Configuration through representation learning (DGM):__ The DGM understands the problem configuration through representation learning. It encodes customer requests—including geographic locations and specific demands—into a global graph embedding. This embedding captures the spatial and demand patterns of customers, enabling the DGM to generate effective depot locations that are well-suited to the distribution of customer needs.
>
>   - __(iii) Providing Flexibility Generation (DGM):__ To achieve flexibility, we generate depots in two distinct modes: direct generation of exact depot locations or production of a multivariate Gaussian distribution for flexible depots sampling. The exact mode ensures precision, when necessary, while the Gaussian mode introduces sampling variability, enhancing the model’s generalization and robustness to diverse customer distributions, especially when regional constraints are present.
>
>   - __(iv) Efficient Evaluation via the Critic Model (MDLRAM):__ To efficiently evaluate the depot sets generated by the DGM, we introduce MDLRAM as a critic model within the DRL framework. MDLRAM provides fast and batch-wise generation of routing solutions based on the depots produced by the DGM. The routing costs computed by MDLRAM act as feedback (or "scores") to the DGM, facilitating its training within a reinforcement learning framework. This feedback loop enables the DGM to iteratively improve depot generation without the need for labeled training data. The objective function, incorporating both depot placement and routing costs, is optimized through this interaction.
>
>   - __(v) Batch-wise computing ability:__ Both the DGM and MDLRAM are designed to process batches of depot sets and routing solutions simultaneously, enhancing scalability and efficiency. During training iterations, the DGM generates a batch of depot sets, which are then evaluated by MDLRAM to generate corresponding routing costs instantly at the same time. This batch-processing capability cannot be efficiently fulfilled by traditional exact or heuristic methods, which typically need to solve each instance individually in case-by-case manner.

---

> > ### Comment · Reviewer_8oe9 · 2024-11-21
> >
> > If using optimization based methods to solve LRP without predefined depot candidates, one can define a larger set of potential locations but associate an additional set of binary variables to these locations, which will take value 1 if we use them as "depot" and 0 if not. This way, it is still possible to use mixed-integer programming models to capture the problem considered in this paper (and if there exist uncertainties, there are literature about how to build two-stage or multistage stochastic integer programming models and how to solve them via branch-and-cut, scenario decomposition, dual decomposition, Bender decomposition methods, etc., to improve the computational results. For sequential decision-making under uncertainty, then you can use Approximate Dynamic Programming, or linear decision rule in a multi-stage stochastic mixed-integer programming models to seek "optimal" policies. So existing optimization based approaches can handle such types of variants of the LRP, and they guarantee solution quality as well as computational efficiency (thanks to the recent advancements in off-the-shelf optimization solvers, such as Gurobi). To make a compelling case that DRL does a better job, benchmark testing and comparison is a must.

---

> > > ### Author Response · Authors · 2024-11-22
> > > **Discussion with Review 8oe9 – Part 2: Regarding the potential exact methods ideas you suggested**
> > >
> > > Thank you for suggesting potential exact methods ideas for addressing the LRP without predefined depot candidates. We appreciate your insights, which could be valuable for future exploration. However, we concern that some of your arguments overlook key aspects of our work's objectives and contributions. Besides, without validation or publication as a recognized literature for LRP without predefined depot candidates, their effectiveness and efficiency remain speculative.
> > >
> > > __1. Regarding the Suggested Potential Exact Methods__
> > >
> > > You proposed that the problem could be addressed by defining a larger set of potential depot locations and associating binary variables with these locations. While this is a valid theoretical direction, we respectfully highlight the following concerns:
> > >
> > > - __(1) No Explicit Published Literature:__ While you mentioned general strategies such as multistage stochastic integer programming, there is no explicit literature that applies these exact methods to the LRP without predefined depot candidates. Existing methods for this specific problem predominantly utilize metaheuristic approaches, which may indicate inherent challenges or constraints in applying exact methods within this context.
> > >
> > > - __(2) Scope of Our Work on DRL-based routing:__ We appreciate the potential of these ideas; however, developing and evaluating such potential exact methods would constitute entirely new research papers themselves. This extends beyond the scope of our current work, which focuses on leveraging Deep Reinforcement Learning to address the LRP without predefined depots.
> > >
> > > __2. Case-Specific vs. Problem-Specific Approaches__
> > >
> > > You mentioned that optimization-based methods could handle variants of the LRP and guarantee solution quality and computational efficiency. We would like to emphasize the following:
> > >
> > > - __(1) Unsubstantiated Assumptions on Efficiency:__ As noted in our paper (Page 2, Lines 58–68), introducing a larger set of potential depot locations significantly increases the problem's dimensionality, leading to scalability challenges. The computational cost of solving such models using exact methods grows exponentially with the problem size, undermining their applicability in scenarios requiring rapid or real-time decision-making. You have suggested several potential solution ideas for LRP without predefined candidates, however, without validation or publication as a recognized literature, their effectiveness and efficiency remain speculative. Demonstrating efficiency in certain VRP variants does not necessarily imply efficiency in the more complex LRP scenarios. Relying on unproven methods does not adequately address the practical limitations we have identified with exact methods for this problem.
> > >
> > > - __(2) Case-Specific Nature:__ Besides, while exact methods can provide optimal solutions for specific instances, our DRL method aims to achieve a higher level understanding on solution patterns—specifically, how customer distributions lead to desirable depot placements and distribution. If solely focusing on deriving a specific optimal solution for a case, exact methods can be used. However, it may not be able to capture underlying distribution patterns.

---

> ### Author Response · Authors · 2024-11-17
> **Response to Reviewer 8oe9 – Part 3**
>
> __[Following Part 2]__
>
> __2. Regarding the results comparison analysis of one component MDLRA:__ Thank you for your insightful questions. We address your concerns as below.
>
> - __(1) Role of MDLRAM and its target on real-world dataset:__ As a critic model within the framework, MDLRAM is crucial due to its ability to rapidly generate high-quality routing solutions in batches, which lays the foundation for depot-generating tasks completed by DGM. *MDLRAM's strengths over traditional methods lie in:*
>   - __(i) Rapid Batch Processing:__ MDLRAM can generate routing solutions for multiple depot sets simultaneously, which is essential for training the DGM efficiently. This capability allows us to evaluate batches of depot sets quickly —a process that would be computationally intensive with traditional methods that typically solve each instance individually in case-by-case manner.
>   - __(ii) Competitive Solution Quality:__ While it may not always match the absolute best solutions from SOTA methods, MDLRAM provides high-quality solutions that consumes significantly less inference time than existing method while close to the optimal. This level of performance is sufficient for effectively training the DGM.
>
> - Therefore, when testing on real-world dataset, our aim is not to establish new SOTA results, instead, since the DGM is trained based on the feedback from MDLRAM, we aim to demonstrate how MDLRAM consumes significantly less inference time than existing method to derive high-quality solutions comparable to BKS, mitigating the potential bias, thereby ensuring the efficacy and robustness for depot generation.
>
> - __(2) Regarding the SOTA Results on real-world dataset:__ we compare our method against four influential LRP literature with benchmark results recognized by the LRP community, using corresponding real-world datasets that are commonly used and typically required for comparison. It is necessary to screen out those less recognized works that arbitrarily add customized constraints to avoid comparing on these real-world datasets, and those works that even not aware the existence of these well-established datasets. This allows us to benchmark our method against well-recognized standards and provides a fair and comparable assessment of its performance.
>
> - Through these comparisons in Table 2, we demonstrate that MDLRAM consumes significantly less inference time than existing methods while deriving high-quality solutions comparable to BKS. This confirms its efficacy for depot generation within our framework.
>
> __3. Comparison with Classical Discrete Optimization Solutions:__ Across the four classic LRP real-world datasets in Table 2, some benchmark results are derived from classical discrete optimization methods, such as the result of B&C in Table 2 which provide exact solution, while some others are derived from heuristic methods, such as HBP and GRASP in Table 2. Besides, for all the cases across the 4 classic LRP real-world datasets, if there are any identified best-known-solution is derived from classical discrete optimization, it is affiliated with an asterisk.
>
> __4. Regarding the potential for applying to large-scale instances:__ Empowered by capabilities such as batch computing, favorable inference time-performance balance, and generalization, the DRL methods for routing problem also have the potential for solving large-scale cases.
>
> - __(1) Framework Design:__ Our framework is designed to be scalable. The neural network models can handle larger datasets by leveraging parallel computing capabilities and batch processing. This allows the model to process multiple instances simultaneously, reducing overall computation time and making it suitable for large-scale applications.
>
> - __(2) Decomposition Techniques:__ We can employ problem decomposition strategies to break down large-scale instances into smaller, more manageable subproblems, which can then be processed in parallel, further enhancing scalability. Previous works have demonstrated the effectiveness of such techniques in improving the scalability of DRL methods (e.g., [7], [8]). By integrating decomposition with our DRL framework, we can extend its applicability to even larger and more complex problem instances.

---

> > ### Comment · Reviewer_8oe9 · 2024-11-21
> >
> > See my comments above regarding what are the issues with the benchmark and computational studies in this paper. The instance sizes (up to 100 demand locations and up to 20 depots) are not considered sufficiently large if compared with what have been solved in the state-of-the-art VRP, LRP literature.

---

> > > ### Author Response · Authors · 2024-11-22
> > > **Discussion with Review 8oe9 – Part 3: Regarding the potential on large-scale instance**
> > >
> > > Thank you for your follow-up comments on large-scale instances. While we appreciate your continued engagement, we are concerned that our previous responses on these points may not have been effectively considered. We would like to briefly recap our responses and further clarify our perspective.
> > >
> > > - __Brief Recap of Our Previous Response regarding dealing large-scale instances:__ In our previous response, we state how existing works can aid DRL method to release the computational ability on large-scale through decomposing and simultaneous computing, supported with specific references [7-8]. We also emphasized that the case-specific nature of exact methods may not always guarantee efficient solution for larger and dynamic problems for all routing scenarios due to their computational demands. In contrast, DRL provides efficient, effective, and generalizable solutions through simultaneously addressing multiple manageable scaled-down cases, which is crucial for dynamic real-world applications.
> > >
> > > - __Instance Sizes and Benchmarking:__ You commented that our instance sizes (up to 100 demand locations and 20 depots) are not sufficiently large. We would like to point out that the instance sizes we use are consistent with those evaluated in recognized LRP real-world dataset. This consistency ensures that our results are directly comparable to existing methods and that our evaluations are adhering to the established standards of the LRP community.
> > >
> > > - __Welcome on Benchmark Recommendations:__ If you are aware of benchmarks in the LRP literature involving significantly large instances that align with this specific problem definition (i.e., LRP without predefined depot candidates), we would greatly appreciate your recommendations and references. It’s worth noting that our focus is on the LRP without predefined depot candidates, not on the Vehicle Routing Problem (VRP) or other related variants. Benchmarks or datasets from VRP studies, while valuable in their context, may not be directly applicable to our specific research focus.

---

> ### Author Response · Authors · 2024-11-17
> **Response to Reviewer 8oe9 – Part 4**
>
> __[Following Part 3]__
>
> __5. Optimality discussion:__ Thank you for raising this insightful discussion, we acknowledge that DRL method does not provide theoretical optimality guarantees in the same way that exact methods do. However, we would like to explain why our approach offers a valuable trade-off between solution quality and computational efficiency, which is particularly beneficial in practical applications where time and resources are limited. We have also made extensive efforts in experimental validation, adhering to evaluation practices in DRL-based routing as established in existing works. Specifically:
>
> - __(1) Justification for the Practical Trade-off between Optimality and Efficiency:__ As discussed earlier regarding the cost-effectiveness balance, when problem scales and complexity continue to increase, exact methods may not offer desirable time efficiency due to its intensive computation and exponential time complexity, even if it has the ability to derive the optimality. In practical environments, such as those with changing demand or real-time constraints, the ability to generate good solutions quickly becomes more important than achieving exact optimality. In such cases, a trade-off between the optimality and efficiency becomes essential, which could be especially significant in the dynamic or uncertain scenarios you kindly mentioned.
>
> - __(2) Promising applicability of our method:__ Building on the advantages discussed in first question, DRL methods are well-suited for real-world scenarios where timely decision-making is critical. Their adaptability and efficiency after training enable the DRL method to:
>   - __(i)__ Consume significantly less time to derive high-quality solution comparable to optimality, which is often more valuable in real-world scenarios where timely decisions are crucial.
>   - __(ii)__ Preserve stable performance on both routing cost and inference time across various problem instances. Our DRL-based method may not guarantee optimality, but it consistently produces high-quality solutions that are close to the optimal, with significantly reduced inference time.
>
> - Therefore, we believe that accepting a trade-off in optimality is worthwhile, as it makes our method more practical for larger problem instances or time-sensitive applications. This trade-off between theoretical optimality and practical efficiency is also a common consideration in machine learning applications for operations research.
>
> - __(3) Adherence to Evaluation Practices in DRL-based routing with extensive experiments:__
>
>   - __(i) Complexity of Theoretical Analysis:__ Due to the stochastic nature of reinforcement learning, the high-dimensional and discrete action spaces, and the absence of convexity or other properties that facilitate theoretical analysis, theoretical optimality analysis for DRL methods in routing problem is less studied, which is evident in current state of research on DRL applied to routing problem [1-10], where studies concentrate on empirical validation through extensive experiments rather than theoretical proofs.
>
>   - __(ii) Focus on Empirical Validation:__ This approach is consistent with existing literature[1-10], where researchers demonstrate the effectiveness of their methods by benchmarking against standard datasets and comparing performance metrics such as solution quality and inference time. We have followed this practice by performing extensive experiments to adhere to this focus on experimental validation common in this DRL-based routing area.
>
>   - __(iii) Our Extensive Experimental Validation:__ Specifically, we have conducted extensive experiments on four real-world datasets to verify that our method approximates optimal routing costs as closely as possible while significantly reducing inference time. Additionally, we conducted experiments on synthetic datasets to assess batch-wise processing capabilities, consistency, stability, and reliability of the solutions. By providing comprehensive experimental evidence, we aim to demonstrate the practical effectiveness and robustness of our approach.
>
> - Our evaluation strategy aligns with the commonly accepted practices in DRL research applied to routing problems, prioritizing the demonstration of effectiveness through experimental results over theoretical analysis due to the reasons mentioned above. By following this approach, we ensure that our work is consistent with the current research landscape[1-10].
>
> - __(4) Potential for Future Theoretical Exploration:__
> We appreciate your thoughtful feedback, which encourages us to explore theoretical aspects in future research. We recognize the significance and aim to keep investigating it. While our method does not offer theoretical optimality, we believe that the extensive experimental validation demonstrates its practical value, offering a worthwhile trade-off between optimality and efficiency, aligning with the current state of research in DRL for routing problems.

---

> > ### Comment · Reviewer_8oe9 · 2024-11-21
> >
> > see my above comments regarding why optimal solutions and theoretical proofs are desired for publishing as a rigorous academic work.

---

> > > ### Author Response · Authors · 2024-11-22
> > > **Discussion with Review 8oe9 – Part 4: Regarding the theoretical guarantees**
> > >
> > > Thank you for your follow-up comments on parts 4. We appreciate your emphasis on theoretical guarantees, but we are concerned that our previous responses may not have been fully considered. We would like to further clarify why theoretical proofs are not the primary focus in DRL-based routing research and how our work aligns with the established evaluation practices in this domain, particularly in top-tier conferences.
> > >
> > > - __Adhering to the established Evaluation Practices in DRL-based routing:__ Due to the stochastic nature of reinforcement learning, the high-dimensional and discrete action spaces, and the absence of convexity or other properties that facilitate theoretical analysis, theoretical optimality analysis for DRL methods in routing problem is less studied, which is evident in current state of research on DRL-based routing [1-10] from top-tier conferences. These studies concentrate on empirical validation through extensive experiments, demonstrating the effectiveness of their methods by benchmarking against standard datasets and comparing performance metrics such as solution quality and inference time. We have followed this practice by performing extensive experiments to adhere to this focus on experimental validation common in this DRL-based routing area.
> > >
> > > - __Extensive Experimental Design from multiple dimension:__ Following the established experimental practices in the DRL-based routing domain, we conducted extensive experiments to demonstrate the practical effectiveness of our method. We evaluated our approach against established benchmarks on real-world datasets, demonstrating its ability to approximate optimal routing costs while achieving significant reductions in inference time. Additionally, we conducted experiments on synthetic datasets to assess batch-wise processing capabilities, consistency, stability, and reliability of the solutions. This multifaceted evaluation provides a comprehensive view of the practical effectiveness and robustness of our approach.

---

> ### Author Response · Authors · 2024-11-17
> **Response to Reviewer 8oe9 – Part 5**
>
> __[Following Part 4]__
>
> We hope this detailed explanation addresses your concerns and demonstrates the practical significance of our approach. Our method offers a favorable balance between solution quality and computational efficiency, making it an alternative to classical methods, especially in dynamic and large-scale scenarios. Thank you again for your engagement and feedback.

---

### Official Review · Reviewer_hQie · 2024-10-30

**Soundness:** 1
**Presentation:** 3
**Contribution:** 2
**Rating:** 5
**Confidence:** 3

**Summary:**

The paper proposes a deep reinforcement learning framework to address the location-routing problem (LRP) for the situation where there are no predefined depot candidates. The framework includes two models: 1. depot generative model (DGM) that generates depots either deterministically or as a multivariate Gaussian distribution for sampling, and 2. multi-depot location-routing attention model (MDLRAM) that   takes the generated depots and the customer requests, and outputs the routing solution to minimize the cost. The authors claim that the main advantage of using the proposed framework lies in the flexibility and the inference speed: the modular framework can be used together or individually, adapted to different scenarios, proactively propose depot placement on the fly, and gives solutions to the routing problem faster.

**Strengths:**

1. Originality: the paper solves a novel LRP problem where there are no predefined depot candidates.
2. Quality: the paper motivates the methodology pretty well. The experimentation and validation of the method seems to be quite rigorous: the paper contains comprehensive evaluation of the framework across different datasets and problems of different scales. It also compares the proposed method with various baselines including exact method.
3. Clarity: the paper provides a clear explanation of the framework. The description of the models is detailed, including structure, objective functions, and training strategy.
4. Significance: the paper has the potential for large impact as it studies a less-studied problem, i.e., LRP without predefined depot candidates.  for the research community, there could be follow-up works on the problem. for real-world applications, the authors also made the case that for use cases including medical rescue and disaster relief, the ability to proactively generate new depot locations is beneficial.

**Weaknesses:**

1. The design of MDLRAM seems to be less supported by the motivation of the paper: the paper stresses the importance of proactively generating depot candidates for flexibility. However, the MDLRAM is a fixed model that takes in a fixed set of customers and depots as input and outputs the routing solution. There is no flexibility provided for changing depots or dynamically planning the routing as the DGM is generating the depots. Considering that the performance of MDLRAM is also inferior to SOTA method, it's unclear what major advantage MDLRAM brings.
2. A potential design flaw in the experiments could be that the evaluation of the DGM model depends on the score given on the MDLRAM model. This could potentially introduce bias since the DGM model is trained to score well for the MDLRAM while the MDLRAM does not provide exact solution or SOTA solution.
3. Randomly generating the depot locations seems to be too weak of a baseline. The heuristics mentioned in the Related Works seems to better candidates.

**Questions:**

1. What is the motivation for MDLRAM? If performance is needed to serve as a good critic model for training DGM, why not use the SOTA method? If flexibility is needed, why not design a more end-to-end framework to connect DGM and MDLRAM and dynamically generate depots and plan routing accordingly.
2. For the evaluation of MDLRAM, for the synthetic data and real-world data settings, different sets of baselines seem to be chosen for comparison. Why is that? The description for the baselines for the synthetic data setting also seems to be lacking. What are ALNS, GA, TS referring to?
3. How would the DGM-Gaussian mode provide more flexibility when compared with the exact mode? Could the authors provide an example of a use case  for the Gaussian mode to motivate it more?

---

> ### Author Response · Authors · 2024-11-17
> **Response to Reviewer hQie - Part 1**
>
> Thank you for recognizing the significance, originality, and potential impact of our work, and also providing thoughtful review. We appreciate your insightful questions and constructive feedback, which have helped us identify areas for improvement. We would like to discuss the concerns and questions in detail below.
>
> __1. Rationale of including MDLRAM as a component within framework:__ Thanks for raising the discussion. We will clarify the role and target of MDLRAM, also explain why it is necessary in our framework. Detailed explanations are also provided in Section 2 “Overview: Chain-of-thought” of the main pages, where we explain how we stack up the framework and the specific functions of each component.
>
> *To summary, MDLRAM is designed to quickly and batch-wisely generate high-quality solution routes based on these depots to serving the customers, as the evaluations for the depot locations generated by the DGM -- a batch-wise task that traditional methods can only finish case-by-case, incurring undesirable inefficiency. Specifically explained as follows:*
>
> - __(1) Necessity of MDLRAM as a Critic Model to evaluates DGM:__ The Depot Generative Model (DGM) aims to generate desirable depot locations based on customer requests. Then training the DGM requires an evaluation mechanism to simultaneously assess the quality of a batch of generated depot sets in each optimization iteration in RL process. This necessitates a model to batch-wisely generating the solution routes based on these depots, with the close-to-the-optimal route cost serving as the evaluations for the depots generated from DGM, thereby facilitating the training of DGM in batch-wise manner. This motivates the introduction of MDLRAM which serves as this critical component by providing fast and batch-wise generation of routing solutions based on the depots produced by the DGM. The routing costs computed by MDLRAM act as feedback (or "scores") to the DGM, facilitating its training in a reinforcement learning framework.
>
> - __(2) Why MDLRAM cannot be replaced by existing SOTA method:__ During each training iteration, the DGM generates a batch of depot sets that need to be evaluated by the critic model to generate corresponding routing costs. However, the SOTA methods for LRP with predefine candidates often involve complex heuristics or exact algorithms that are computationally intensive and not designed for batch processing, which typically need to solve instances case-by-case for generating a batch of results, each consuming a non-negligible inference time, significantly slowing down the training process. This computational intensity poses a challenge for integrating them into a reinforcement learning framework that requires rapid and repeated evaluations.
>
> - __(3) How to help MDLRAM avoid bias during training DGM:__  We also recognize the concern regarding potential bias since the DGM is trained based on the feedback from MDLRAM. Because MDLRAM serves as a critic model for DGM, it should be robust enough to provide a reliable score for assessing the generated depots from DGM. That means, the score is expected to solely reflect the quality of the generated depots, ruling out the influence of LRP routing solution’s quality as much as possible. *To address this, we have made following three aspects of efforts:*
>   - __(i) Pretraining MDLRAM:__ We pretrain MDLRAM separately to approximate optimal routing costs as closely as possible. This involves training MDLRAM on a large set of instances to ensure it provides reliable solution routes as evaluations.
>   - __(ii) Fixing MDLRAM During DGM Training:__ Once pretrained, MDLRAM's parameters are fixed during the training of the DGM. This ensures that improvements in routing costs are attributable to better depot locations generated by the DGM, rather than changes in MDLRAM's routing performance, thereby facilitating a robust training process for DGM.
>   - __(iii) Validation Against Benchmarks:__ We validate MDLRAM's outputs against benchmark methods to ensure its reliability as a critic model. By comparing its performance with established methods, we confirm that MDLRAM provides accurate and trustworthy evaluations for the depot sets generated by the DGM.

---

> ### Author Response · Authors · 2024-11-17
> **Response to Reviewer hQie - Part 2**
>
> __[following part 1]__
>
> - __(4) Why not dynamically generate depots and plan routing accordingly:__ It is precisely our consideration of this question that motivates our decision to pretrain MDLRAM and set it fixed during training DGM. We’re delighted to share the same consideration with you on this matter. The framework consists of two components. As mentioned above, the training strategy we choose is pretraining MDLRAM to minimize the routing cost, aiming to facilitate an robust and unbiased evaluation for DGM. Once MDLRAM is fixed, during DGM's training, further reduction in costs is basically contributed by DGM’s optimized ability on identifying better depots. *However, if directly training these two models in a joint mode, undesirable challenges will be posed in three aspects:*
>   - __(i) Training Instability:__ Joint training could lead to instability due to simultaneous optimization of both DGM and MDLRAM. The MDLRAM generates solution based on Markov Decision Process (MDP), relying on a stable reinforcement learning process. The changes of DGM’s network parameters could adversely affect MDLRAM’s convergence and overall training stability.
>   - __(ii) Variable Control:__ Separating the training of DGM and MDLRAM allows us to control variables more effectively. By fixing the well-trained MDLRAM, we aim to attribute the improvements in the total cost during DGM’s training to better depot generation by the DGM, rather than the intertwined effects from both models’ simultaneous learning.
>   - __(iii) Flexibility of Individual Components:__ If joint training, the reduction of the total cost is applied to optimize both DGM and MDLRAM, their performance might be tightly coupled, making it difficult to utilize them separately without retraining. Training the models separately enhances the flexibility of the framework. MDLRAM and DGM can be used independently in other contexts if desired.

---

> ### Author Response · Authors · 2024-11-17
> **Response to Reviewer hQie - Part 3**
>
> __[following part 2]__
>
> __2. Performance of MDLRAM compared to baseline methods on both real-world dataset and synthetic dataset:__
> - __(1) Role of MDLRAM as a Critic Model and its target on real-world dataset:__ MDLRAM's primary role is to serve as an efficient and reliable critic for the DGM during training, rather than to establish new state-of-the-art results in routing. Its strengths lie in:
>   - __(i) Rapid Batch Processing:__ MDLRAM can generate routing solutions for multiple depot sets simultaneously, which is essential for training the DGM efficiently. This capability allows us to evaluate a batch of depot sets quickly, which would be computationally intensive with traditional methods.
>   - __(ii) Competitive Solution Quality:__ While it may not always match the absolute best solutions from SOTA methods, MDLRAM provides high-quality solutions that consumes significantly less inference time than existing method, which are sufficient for effectively training the DGM.
>
> - Therefore, when testing on real-world dataset, our aim is not to establish new SOTA results, instead, since the DGM is trained based on the feedback from MDLRAM, we aim to demonstrate how MDLRAM consumes significantly less inference time than existing method to derive high-quality solutions comparable to BKS, mitigating the potential bias, thereby ensuring the efficacy and robustness for depot generation.
>
> - __(2) Comparisons on Real-world datasets and Synthetic Datasets:__ While the primary goal of MDLRAM is to serve as a robust critic model for the DGM, rather than achieving SOTA results for the traditional scenario, we performed comparisons on this traditional configuration to demonstrate its robustness and effectiveness.
>
>   - __(i) Recognized Benchmarks on Established Real-world Dataset:__ we compare our method with four influential LRP literature with benchmark results recognized by the LRP community, using corresponding real-world datasets that are commonly used and typically required for comparison. It is necessary to screen out those less recognized works that arbitrarily add customized constraints to avoid comparing on these real-world datasets, and those works that even not aware the existence of these well-established datasets. This allows us to benchmark our method against well-recognized standards and provides a fair and comparable assessment of its performance.
>
>   - __(ii) Rationale for baselines selection:__ The SOTA methods, which only presents results on real-world dataset, do not reveal publicly available source code or detailed parameters setting for reproduction, making it not suitable to yield comparable results on the synthetic dataset, that’s why we choose to detailedly compare our MDLRAM with SOTA methods on their real-world dataset, which highlights not only the solution quality and less inference time, but also the generalization performance across various customer distribution.
>
>   - As for the Synthetic Dataset, it is typically used to assess MDLRAM's robustness and stability across a variety of instances across a batch of results through monitoring metrics such as error bars (deviation), to demonstrate how the DRL model can achieve consistent and reliable performance on cases. But to also facilitate a comprehensive comparison and complement the limitations of SOTA methods on synthetic data, we include results from well-known, widely used, and publicly available heuristic methods in routing tasks, including ALNS (Adaptive Large Neighborhood Search), GA (Genetic Algorithm), and TS (Tabu Search), making them standard benchmarks for synthetic data experiments as suitable baselines.

---

> ### Author Response · Authors · 2024-11-17
> **Response to Reviewer hQie - Part 4**
>
> __[following part 3]__
>
> __3. Choice of Baselines for Depot Generation:__ Regarding why choosing Randomly generating the depot locations as a baseline in depot generation comparison, we'd like to clarify our rationale from the following perspectives:
>
>   - __(1) Constraints to single-depot/up-to-2 depots for existing heuristic methods:__ The heuristics mentioned in related work are still under the constraints of single-depot or up-to-two depots, and may not generalize well in our problem setting, which involves generating multiple depots without predefined candidates. As such, they cannot provide a reliable and comparable results for the comparison analysis.
>
>   - __(2) Validity of Random Sampling as a Baseline:__ Random depot generation serves as a valid and meaningful baseline for our problem due to several reasons:
>     - __(i) Efficiency in Evaluation:__ Leveraging the batch processing capabilities of the MDLRAM, we can efficiently evaluate a large number of randomly generated depot sets. This provides a robust comparison while keeping computational costs manageable.
>     - __(ii) Benchmarking Improvement:__ Since the quality of depots, whether generated by random sampling or the DGM, is evaluated using the MDLRAM, the random sampling establishes a baseline performance level, allowing us to demonstrate the effectiveness of our DGM in identifying better depot locations.
>
>   - __(3) Ablation Study:__ Including random generation in our experiments also serves as an ablation study, highlighting the contribution of the DGM in improving routing costs over naive depot selection.
>
> __4. Flexibility Provided by DGM's Gaussian Mode:__ In response to the question about how the DGM-Gaussian mode provides more flexibility compared to the exact mode, we would like to elaborate on following aspects, supplemented by use case example:
>
>   - __(1) Flexibility Advantage:__ Among the two generation modes for DGM, compared with the exact mode, which generates precise depot locations based on customer requests, the Gaussian mode offers additional benefits from:
>     - __(i) Probabilistic Depot Generation:__ The Gaussian mode allows the DGM to generate a distribution over potential depot locations rather than fixed points. This is particularly useful in scenarios where depot placement is subject to uncertainty or variability, enabling the model to account for fluctuations and adapt to dynamic conditions.
>     - __(ii) Adaptability to Constraints:__ It enables sampling from regions that satisfy specific constraints, such as avoiding restricted areas or adhering to regional regulations. This adaptability ensures that the generated depots remain feasible and practical within the given operational environment.
>
>  - __(2) Use Case Example with Visualized Comparison in Figure 4:__ Consider disaster relief operations where depot locations must be established rapidly and may need to change due to evolving conditions like accessibility, safety concerns, or shifting demand. The Gaussian distribution enables sampling of depot locations within a probabilistic region, offering the necessary adaptability to respond to such dynamic environments. To illustrate this flexibility, Figure 4 visualizes how the exact depot locations and the Gaussian distributions present over the same set of customer requests. This helps to intuitively demonstrate the flexibility of the Gaussian mode.
>
> We hope that our detailed explanations address your concerns and clarify the motivations and advantages of our framework components. We believe that the MDLRAM is essential for efficiently training the DGM and that the flexibility offered by the Gaussian mode of the DGM enhances the applicability of our method to dynamic and uncertain environments. Thank you again for your valuable time, constructive feedback and consideration, which has helped us enhance the presentation and understanding of the key aspects of our work.

---

> > ### Comment · Reviewer_hQie · 2024-11-27
> >
> > Thanks the authors for the clarifications. I see the point that doing training, fast inference is needed so that MDLRAM might be better suited than SOTA methods. What about at evaluation time for the DGM? Why is MDLRAM a better choice than other methods there?

---

> > > ### Author Response · Authors · 2024-11-27
> > > **Discussion with Reviewer hQie - Part 1**
> > >
> > > Thank you for your engagement and taking the time to read through our clarifications. We appreciate your recognition of the necessity of MDLRAM during training due to its efficient batch-wise inference capabilities, while also our efforts in verifying MDLRAM as a critic model. We thank you for your follow-up question.
> > >
> > > You raise an important point regarding the evaluation phase of the DGM--Why MDLRAM is a better choice than other methods there, providing us the opportunity to further explain the necessity of MDLRAM during evaluation. We will discuss this from following key points in detail.
> > >
> > > __1. Consistency Between Training and Evaluation__
> > >
> > > __Alignment of Evaluation Metrics:__ Using MDLRAM during both training and evaluation ensures consistency in how the depot locations/distribution generated by the DGM are assessed. Since the DGM is trained to minimize routing costs as evaluated by MDLRAM, it makes sense to use the same evaluation method during testing to accurately measure the improvements learned during training.
> > >
> > > This alignment ensures that the performance improvements learned during training are accurately measured during evaluation, avoiding any discrepancies that might arise from using different downstream routing solvers. Consistent evaluation metrics provide a reliable basis for assessing the DGM's effectiveness.
> > >
> > > __2. Computational Efficiency Remains Important During Evaluation__
> > >
> > > Even during evaluation, computational efficiency is still crucial, especially when dealing with large datasets or specific application mode. MDLRAM's ability to rapidly generate close-to-optimal routing solutions for multiple cases simultaneously continues to provide desirable advantages in terms of inference time.
> > >
> > > Let’s see what specifically happens during DGM’s evaluation. The following factors necessitate the use of MDLRAM during evaluation to avoid the case-by-case solving manner of existing methods:
> > >
> > > - __(1) Handling Large Numbers of Testing Cases:__ Evaluating the DGM involves assessing its performance across a large number of instances. Traditional methods, which operate on a case-by-case basis, become inefficient and impractical in this —especially considering the fluctuations in inference time for different cases, as reflected in Table 2. MDLRAM's batch-wise processing enables efficient evaluation of multiple cases simultaneously, saving valuable time and computational resources.
> > >
> > > - __(2) Support for the Gaussian Distribution Mode:__ The Gaussian mode allows the DGM to generate a distribution over potential depot locations rather than fixed points. When evaluating the DGM's Gaussian distribution mode, multiple samples of depots set are drawn to capture the performance of the generated distribution, reporting both the best identified depots solution from the distribution and the average quality of the depots solution from the distribution. Evaluating these samples also requires efficient simultaneous processing. Compared with existing methods case-by-case manner, MDLRAM's ability to handle batch evaluations allows for quick assessment of these samples without incurring high computational costs.
> > >
> > > To unlock the full potential of the DGM and fully realize the benefits of its distribution mode—as well as to evaluate its performance across diverse conditions—an efficient evaluation method like MDLRAM is necessary. This is especially important in real-life scenarios where rapid assessments are needed to make timely decisions.
> > >
> > > __3. Providing Robust Solving Performance__
> > >
> > > MDLRAM consistently delivers robust performance in terms of both inference time and close-to-optimal solution quality across different cases, without the need for parameter adjustments for specific instances. This is a notable advantage over some existing methods. Specifically:
> > >
> > > - __(1) Avoiding case-specific Parameter Tuning:__ Traditional heuristic methods may achieve the desirable result on individual cases through case-specific parameter tuning. However, parameters optimized for one instance may not generalize well to others, necessitating manual adjustments. In contrast, MDLRAM provides high-quality solutions without the need for such tuning.
> > >
> > > - __(2) Stable Solving Performance Across Cases:__ MDLRAM demonstrates stable solving performance across different cases. As shown in our experiments, it achieves high-quality solutions comparable to those obtained by existing methods, while notably reducing inference time. More importantly, the stable performance can be maintained across various instances, ensuring that the evaluation results are reliable and reflective of the DGM's true capabilities.

---

> > > ### Author Response · Authors · 2024-11-27
> > > **Discussion with Reviewer hQie - Part 2**
> > >
> > > __[Following Discussion with Reviewer hQie - Part 1]__
> > >
> > > __4. As a Joint Use Case Demonstrating the Framework’s Flexibility and Adaptability in Practical Usage__
> > >
> > > Additionally, we would like to discuss how the involving of MDLRAM in DGM’s evaluation provides the complete workflow of a use case, showing the framework’s flexibility and adaptability in practical usage. Beyond serving as a validation, the evaluation process also need to reflect how the framework can be effectively used in potential practical scenarios. The involving of MDLRAM enhances the practicality of our framework in real-world applications, especially those with limited computational resources or strict time constraints.
> > >
> > > - __(1) Handling Dynamic Scenarios:__ In environments where conditions change rapidly, the seamless integration of MDLRAM with DGM allows for quick re-evaluation of the generated depot configurations. This enables the system to adapt swiftly to new information, which is crucial in dynamic settings.
> > >
> > > - __(2) Facilitating Iterative Improvements:__ If further adjustments or iterative evolution on the depots set generated from DGM are needed during deployment, MDLRAM enables rapid assessments without significant delays, supporting agile decision-making processes.
> > >
> > > We appreciate the opportunity to share these thoughts with you and to further explain the necessity of MDLRAM during the DGM's evaluation phase, from both evaluation and practical usage perspectives. We hope that our detailed explanations address your concerns and provide a clearer understanding of our framework's design choices.
> > >
> > > These insightful questions have been helping us enhance the logic and clarity of our work. We are grateful for your continued engagement and constructive feedback, and we are delighted to be part of this informative discussion. Thank you once again for your time and consideration.

---

### Official Review · Reviewer_U1kY · 2024-10-30

**Soundness:** 3
**Presentation:** 2
**Contribution:** 3
**Rating:** 8
**Confidence:** 4

**Summary:**

This paper proposes a DLR approach for the Location-Routing Problem that generates possible depot locations either in an exact way (by proposing exact locations for the depots) or in a probabilistic way (by providing a Gaussian probability distribution where depot locations can be efficiently sampled). The DLR approach employs a multi-head attention model as a critic model which is capable of assessing the depots generated by the generative model. The approach is tested extensively on synthetic and real data and compared to other approaches in the literature.

**Strengths:**

The paper is very dense (probably conditions by the page limits) but the authors make a rather good job in providing a clear overview and a good discussion. The quality of the paper is high, as the experiments and the results shown are able to capture a wide range of the implications of the approach. The experiments are extensive and all important details and parameters are provided.
The novelty of the presented approach is mainly driven by the generative approach used for generating depot locations alongside the routes for each problem instance. The authors do overall a good job in motivating the novelty of their contributions.

**Weaknesses:**

Although the appendix provides the pseudocode for the main components of their approach, source code is missing. This could hinder the ability to reproduce the results obtained.
The authors, probably due to the page limits, seem to be forced to reduce the size of the tables and the figures in the main text considerably, which negatively affects the quality of the exposition.
In general, one cannot dismiss the impression that the authors are not only providing technical details in the appendices, but also including content that would have gone into the main text were it not the page limits. This impression gets reinforced specifically in B.4 where authors address limitations and future work, two topics that typically included into the main text and not in the appendices. In the introduction, there is no overview of the structure of the paper as usual.

**Questions:**

- In Table 1 and Table 3 what is the meaning of boldface and asterisk? It cannot be best result so far, as e.g. in Table 3 different results for the same row are marked with asterisk. The authors could better explain this in the caption or in the main text.
- Figure 2 is rather small, it would be helpful to increase size if page limits allow for this.
- I strongly recommend to include a link to publicly available source code to increase the reproducibility of the study.

---

> ### Author Response · Authors · 2024-11-17
> **Response to Reviewer U1kY**
>
> Thank you for your positive feedback on our work’s contribution and providing the constructive suggestions. We are delighted that you find our work valuable, and we appreciate your helpful insights on how to improve both the presentation and content of our paper. Here are the specific actions and corresponding clarifications.
>
> __1. Presentation and Content Improvements:__ We appreciate your understanding of the constraints posed by the page limits and our efforts to present the main details effectively. To enhance the presentation and layout of the paper, we plan to implement the following actions:
> - __(1) Integrating Limitations and Future Work into the Main Text from Appendix:__ In the revised version, we will integrate a summary of these topics into the conclusion section, while directing readers to the appendix for more detailed discussions.
> - __(2) Optimizing Table Sizes and Readability:__ Because we aim to present comprehensive analysis for each component while both on real-life dataset and synthetic dataset, the presentation is a somewhat tight. To address this, we will reallocate space within the main text by streamlining some of the experimental setup descriptions and moving additional details to the appendix, thereby allowing more space for tables.
> - __(3) Enhancing Figure 2:__ We agree that Figure 2, which visually display the training process of two modes, is currently small and may be difficult to read. In the final version, we will simplify the description and enlarge the figure, improving its clarity.
> - __(4) Including paper structure overview:__ Thanks for noting the need of a paper structure overview in the introduction. We will include a brief outline to guide readers through the organization of content.
>
> __2. Clarification on Table Annotations:__ We apologize for omitting the meaning of boldface and asterisk in Table 1 and Table 3, we will clarify this in the captions to enhance readability. The meaning is also described here for clarification:
>
> - __(1) In Table 1:__ Boldface and asterisks highlight the best performance achieved by MDLRAM using the sampling test strategy when evaluated on synthetic datasets. This result outperforms both its greedy test strategy and other heuristic methods.
> - __(2) In Table 3:__ Boldface and asterisks indicate that, for the corresponding scale and test strategy, whether the DGM (with its two generation modes) achieves better results or random sampling identify better results. Specifically, when the DGM achieves better results, the asterisk denotes which mode of the DGM—either the exact depot locations or the multivariate Gaussian distribution—provides the superior performance.
>
> __3. Regarding the Source Code Availability:__ We understand the importance of reproducibility. We are in the process of preparing our code for public release, incorporating improvements based on feedback received during the review process. We will include a link to the repository in the final version of the paper if accepted, enabling related researchers to replicate our results and build upon our work.
>
> We appreciate your thoughtful feedback, which has been significant in improving our paper,  enhancing both the readability and the impact. We are committed to addressing all the points you've raised and believe that the revisions will strengthen the paper. Thank you once again for your valuable time and constructive comments.

---

> > ### Comment · Reviewer_U1kY · 2024-11-26
> >
> > Thank you very much for your clarifications and for addressing all of my comments and suggestions.
> > Since my score was already accept, I won't change the score.

---

### Official Review · Reviewer_B5p6 · 2024-11-09

**Soundness:** 2
**Presentation:** 1
**Contribution:** 1
**Rating:** 1
**Confidence:** 5

**Summary:**

This paper focuses on a DRL framework for location routing problems in which potential depot locations are not fixed a priori but can be freely determined in a continuous plane. The authors propose a generative DRL approach that generates potential depots and subsequently derives an LRP solution.

**Strengths:**

see weaknesses

**Weaknesses:**

I am afraid to write such a negative review, which will clearly be a disappointment for the authors, but in my opinion this paper should not be published for the reasons outlined below. I hope that my concerns will still help the authors to refine their research in the future.

1) As the authors point out in the literature review part of their work, research on solving LRPs without predefined depot locations is very scarce. From my experience in academia and practice, there is a good reason why this research is scarce: solving an LRP without predefined depot locations is a problem that is hardly of interest to the underlying applications.
The depots which are placed when solving an LRP are typically warehouses or logistics hubs. Suitable locations for such buildings are usually sparse, which is why there always exists a predefined discrete set of potential locations, which makes the problem proposed by the authors obsolete.
If one wants to identify suitable regions for depots, one usually solves a facility location problem and uses a cost approximation for routing costs as in such a setting where the problem is solved in a continuous plane, the error made by placing the depot locations at points that are not available as a location is high and outweighs a potential accuracy win on getting accurate routing costs.

2) While I believe that utilizing DRL for vehicle routing problems is still in its infancy and holds potential for many promising research contributions, I doubt that aiming to solve strategic planning problems with similar methodologies for two reasons: first, when solving a strategic problem the sunk costs are very high. In this case, one favors solution quality and accuracy, realized in classical combinatorial or robust optimization approaches, over fast decision-making, which is the main advantage of a DRL approach. Second, while one can easily create many training instances for an operational problem by collecting daily or hourly data, this is different for a strategic problem that is solved once every 10 to 20 years. I do not see any reasonable way to create the 1,280,000 instances the authors require for training.

Ignoring the unrealistic training process, a superior solution quality could justify the method proposed by the authors from a purely academic perspective. However, the reported solution quality is inferior to that of existing classical approaches. Here it might be worth mentioning that the reference solutions claimed to be the state of the art by the authors are roughly 15 years old. Many papers have improved upon these so that the gap between the proposed methodology and the actual state of the art is much larger.

Based on these main concerns, I see neither a pracitcal nor an academic value in the work proposed and recommend to not publish it.

**Questions:**

see weaknesses

---

> ### Author Response · Authors · 2024-11-17
> **Response to Reviewer B5p6 - Part 1**
>
> Thank you for taking the time to review our paper. We appreciate the sharing of your experience and would like to discuss the concerns you've raised point-by-point.
>
> __1. Regarding the scarcity of related works:__ Thanks for the experience that the scarce number of related works indicate a problem is hardly of interest. Regarding this, we would like to respectfully raise some points-of-view to discuss why this does not apply to our problem scenario, while also why our research presents significance.
>
> - __(1) Scarcity of related works resulting from constraints:__ There may be multiple reasons leading to a scarcity of related works, such as unsolved constraints or unidentified problems, which we will detailedly discuss in following response. We do not totally exclude the reviewer’s concern, but we also believe this presents an opportunity and motivation for researchers to explore and identify new application scenarios, thereby expanding and contributing to the existing research landscape. Rather than focusing solely on improving state-of-the-art results on the most basic well-studied problems, exploring new and potential problems can yield significant advancements. We believe this also aligns with ICLR's perspective on encouraging novel research directions.
>
> - __(2) Significance of LRP without predefined depots through review of existing works:__ In route planning domain, numerous variants integrate various constraints and objectives from customers, fleet owners, and regulators, and are addressed by various types of methods. Each variant corresponds to different application scenarios with specific requirements and operational targets. We believe that potential applications have their own significance, even if they have not yet been identified or extensively studied yet. Identifying and addressing these applications is a crucial task for researchers and practitioners.
>
> - As mentioned in the introduction, through investigating existing works (Page3-Line127), we identified that the LRP without predefined depot locations remains a less-explored area with substantial practical relevance, specifically for those scenarios that require rapid depot establishment and adjustment, such as medical rescue and disaster relief operations. Furthermore, through reviewing these existing works, we identify the unsolved efficiency constraints which can be addressed by DRL’s generative capability. We aim to broaden the scope of LRP research and provide solutions for scenarios that are not adequately addressed by existing methods. Specifically:
>
>   - __(i) Practical Applications Requiring Rapid Depot Establishment:__ As detailed in the introduction part, there are critical real-world situations where the assumption of predefined depot locations is not feasible. Examples include emergency medical rescue and disaster relief operations, where infrastructure may be compromised or non-existent. In such cases, there is an urgent need to determine or adjust depot locations quickly and efficiently to facilitate effective response efforts.
>
>   - __(ii) Constraint of Computational Intensity:__ Regarding this extended LRP scenario without predefined depot candidates, the existing methods predominantly employing heuristic strategies for attempting new depots across a planar area. In this situation, if simply expanding the search of the depot candidates across the map and aimlessly attempting new points, then undesired increasing on problem scale will be incurred, thereby leading to excessive time consumption and expensive computation. Therefore, devising a method to proactively generate high-quality depots, satisfying the depot location constraints for LRP scenario without predefined depot candidates, is well-motivated. Our approach addresses this need by efficiently generating depot locations without exhaustive searching.

---

> ### Author Response · Authors · 2024-11-17
> **Response to Reviewer B5p6 - Part 2**
>
> __[Following Part 1]__
>
> __2. How DRL method contributes to addressing the constraints of existing related works:__ Our DRL method consists of DGM and MDLRAM. Adhering to this architecture, we will delineate how our DRL method contributes to advancing LRP without predefined depots candidates from following aspects:
> - __(1) Reducing Computational intensity with proactive generation:__ Existing methods predominantly employ heuristic strategies that attempt depot placements across the map. This approach can lead to an undesirable increase in problem scale, resulting in excessive time consumption and computational expense due to the need to evaluate numerous potential depot locations individually. Our approach leverages the generative capabilities of DRL to proactively generate high-quality depot locations. By doing so, we avoid repetitive and aimless searching, reducing computational complexity and improving efficiency. The DGM generates depot locations based on learned representations of customer demand, streamlining the process and focusing on promising areas.
> - __(2) Understanding Problem Configuration through representation learning (DGM):__ The DGM understands the problem configuration through representation learning. It encodes customer requests—including geographic locations and specific demands—into a global graph embedding. This embedding captures the spatial and demand patterns of customers, enabling the DGM to generate effective depot locations that are well-suited to the distribution of customer needs
> - __(3) Providing Flexibility (DGM):__ To achieve flexibility, we generate depots in two distinct modes: direct generation of exact depot locations or production of a multivariate Gaussian distribution for flexible depots sampling. The exact mode ensures precision, when necessary, while the Gaussian mode introduces sampling variability, enhancing the model’s generalization and robustness to diverse customer distributions, especially when regional constraints are present.
> - __(4) Efficient Evaluation via the Critic Model (MDLRAM):__ To efficiently evaluate the depot sets generated by the DGM, we introduce MDLRAM as a critic model within the DRL framework. MDLRAM provides fast and batch-wise generation of routing solutions based on the depots produced by the DGM. The routing costs computed by MDLRAM act as feedback (or "scores") to the DGM, facilitating its training within a reinforcement learning framework. This feedback loop enables the DGM to iteratively improve depot generation without the need for labeled training data. The objective function, incorporating both depot placement and routing costs, is optimized through this interaction.
> - __(5) Batch-wise computing ability:__ Both the DGM and MDLRAM are designed to process batches of depot sets and routing solutions simultaneously, enhancing scalability and efficiency. During training iterations, the DGM generates a batch of depot sets, which are then evaluated by MDLRAM to generate corresponding routing costs instantly at the same time. This batch-processing capability cannot be efficiently fulfilled by traditional exact or heuristic methods, which typically need to solve each instance individually in case-by-case manner
> - __(6) Additional advantage of Applicability to Traditional Scenarios (MDLRAM):__ For traditional scenarios with predefined candidates, we also demonstrate that the MDLRAM component can be used individually within this configuration to select depots from pre-given candidates, which is an additional advantage of MDLRAM, empowered by its role as a critic model for generating solution routes to evaluate the depots generated by DGM within the entire framework.
>
> __3. Regarding the concern about applying DRL methods to strategic problems:__ we would like to provide a perspective about how our approach can enhance even the traditional scenario with predefined depot candidates.
>
> In traditional scenarios where depot candidates are predefined, how these candidates are initialized is also crucial for the quality of the final solution, yet determining these candidates is not always straightforward. Even if the exact methods may derive the optimal solution, it still hinges on the quality of these initial candidates.
>
> In such cases, our approach addresses this challenge effectively. Even if the exact locations generated by the DGM’s exact mode are not directly utilized, the DGM’s Gaussian mode can initialize a more informed and strategic set of depot candidates. Then, if necessary, flexibly using other methods to selected from these candidates can also offer signify a valuable contribution to the overall solution process.  This dual functionality is precisely why we set DGM with both exact mode and gaussian distribution mode.

---

> ### Author Response · Authors · 2024-11-17
> **Response to Reviewer B5p6 - Part 3**
>
> __[Following Part 2]__
>
> __4. A discussion on higher level advantages of using DRL method:__ In our previous response, we discussed how DRL specifically advances the LRP without predefined depots. Here, we would like to delve deeper into our motivation for employing DRL by discussing the cost-effectiveness balance among different methods.
>
> - __(1) Balancing Pros and Cons of Different Methods:__ Each type of method—exact algorithms, heuristics, and DRL—has its advantages and limitations, which is why we consider the "cost-effectiveness" balance when comparing methods for routing problems. For different routing variants with varying levels of complexity, this cost-effectiveness balance consideration also changes. This motivates new methods like DRL come into view, following the emergence of more complicated routing task.
>
> - __(2) Expanding Cost-Effectiveness Considerations:__ Traditionally, considerations of cost-effectiveness include computational cost, inference time, and routing performance. This was also how the heuristic methods emerged to complement the intensive computation of exact methods when dealing with more complex configurations. Now, beyond these factors, as problem scales and complexity continue to increase, factors like batch-wise computing and generality (i.e., reduced effort required to adapt existing methods to new scenarios) also become important in the cost-effectiveness balance. This is where DRL offers significant advantages, help to alleviate two dilemmas faced by traditional methods:
>   - __(i) Repetitive Search Processes:__ Traditional methods typically require initiating a new search procedure for each individual case, even with minor alterations, leading to a case-by-case solving approach. This is not desirable in dynamic scenario with large problem scales.
>   - __(ii) Adaptability to Variants:__ Expanding a specific VRP variant algorithm to accommodate other specialized variants with different task configurations often lacks adaptability in traditional methods.
>
> - __(3) Key Benefits of DRL:__ We certainly appreciate the strengths of exact and heuristic methods. However, when certain more important advantages are difficult to achieve with traditional methods, DRL becomes a valuable alternative to attain a more desirable cost-effectiveness balance. The key advantages of DRL present in following points:
>   - __(i) Batch-wise Computing:__ DRL models can process multiple instances simultaneously, improving computational efficiency and making them suitable for large-scale problems and dynamic scenario.
>   - __(ii) Learning Desirable Solution Patterns through Sequential Decision-Making Framework:__ The Markov Decision Process (MDP) framework enables DRL models to capture problem configurations and sequentially generate solutions, allowing more flexible adaptation to various routing variants.
>   - __(iii) Stable Performance After Training:__ Once trained, the model can solve new cases with same task configurations with stable performance, achieving a good balance between inference time and solution quality.
>
> - __(4) Addressing "Sunk Costs":__ We understand that the "sunk costs" mentioned may refer to the training costs associated with DRL. However, we believe this is a reasonable trade-off for the benefits outlined above, offering a favorable cost-effectiveness balance compared to traditional methods, especially in complex and dynamic scenarios, which are the targeted application areas we have discussed in introduction section.
>
> - Moreover, in dynamic scenarios where we want to achieve simultaneous computing to avoid initiating a new search process for each new case with minor alterations, DRL serves as a strong substitute for traditional methods. In this sense, investing hours of training to achieve instant, high-quality, batch-wise solving, means that the "sunk cost" for DRL may even be lower than that of traditional methods over time.

---

> ### Author Response · Authors · 2024-11-17
> **Response to Reviewer B5p6 - Part 4**
>
> __[Following Part 3]__
>
>
> __5. Regarding the Results Comparison of the MDLRAM Component within Our Framework:__ We appreciate your insights regarding the analysis of MDLRAM within our framework. We would like to clarify the role of MDLRAM and the arrangement of our comparisons on real-world datasets.
>
> - __(1) Role of MDLRAM as a Critic Model and its target on real-world dataset:__ As mentioned in the “Overview: Chain-of-Thought” section, during training, MDLRAM serves as a critic model for batch-wise generation of solution routes, as evaluations for the depots generated by the DGM, facilitating the training of DGM efficiently in a batch-wise manner.
>
> - As a critic model within the framework, MDLRAM stands out for its rapidity to plan high-quality solutions in batches, which lays the foundation for depot-generating tasks completed by DGM. Therefore, when testing on real-world dataset, our aim is not to establish new SOTA results, instead, since the DGM is trained based on the feedback from MDLRAM, we aim to demonstrate how MDLRAM consumes significantly less inference time than existing method to derive high-quality solutions comparable to BKS, mitigating the potential bias, thereby ensuring its efficacy and robustness for depot generation.
>
> - __(2) Comparison with SOTA Results on real-world dataset:__ While the primary goal of MDLRAM is to serve as a robust critic model for the DGM, rather than achieving SOTA results for the traditional scenario, we performed comparisons on this traditional configuration to demonstrate its robustness and effectiveness.
>
> - We compare our method with SOTA results documented in four influential LRP literature acknowledged by the LRP community, using corresponding real-world datasets that are commonly used and typically required for comparison. It is necessary to screen out those less recognized works that arbitrarily add customized constraints to avoid comparing on these real-world datasets, and those works that even not aware the existence of these well-established datasets. This allows us to benchmark our method against recognized standards and provides a fair and comparable assessment of its performance.
>
> - Through these comparisons in Table 2, we demonstrate that MDLRAM consumes significantly less inference time than existing methods while deriving high-quality solutions comparable to BKS. This confirms its efficacy for depot generation within our framework.
>
> We hope that our clarifications address your concerns and illustrate the motivation and significance of our work. Our goal is to extend the research area by identifying new application scenarios and providing new insights into critical applications where traditional methods may fall short. We welcome any further questions or suggestions you may have and are eager to engage in a constructive discussion to improve our work. Thank you again for your feedback.

---

> > ### Comment · Reviewer_B5p6 · 2024-11-22
> >
> > thank you for the clarifications, my score though remains unchanged

---

### Meta-Review · Area_Chair_fcPt · 2024-12-20

**Metareview:**

This paper proposed a learning based method to generate candidate depot in the location-routing problem. It can operate in two modes, i.e. direct generation of exact depot locations and the creation of a multivariate Gaussian distribution for flexible depots sampling. Some reviewers found the proposal interesting, but three out of four reviews are negative. The key weaknesses are: 1) lacking a good motivation of generating depots in practical scenarios and using DRL for strategic planning; 2) lacking detailed discussion w.r.t location-routing problems; and 3) lacking thorough evaluation against SOTA operational research methods. I agree that all the three points are valid, and the paper requires substantial improvement to address them. Therefore, I recommend rejection.

**Additional Comments On Reviewer Discussion:**

Authors provided detailed point-to-point respones to reviewers' comments. However, most of their concerns remain. Most of authors' responses are clarification, and no further results are provided. It appears that the authors' responses are not strong enough to address reviewers' concerns.

---

### Decision · Program_Chairs · 2025-01-22

Reject